

# Improving the Mean and Uncertainty of Ultraviolet Multi-Filter Rotating Shadowband Radiometer In-Situ Calibration Factors: Utilizing Gaussian Process Regression with a New Method to Estimate Dynamic Input Uncertainty

Maosi Chen [1], Zhibin Sun [1], John M. Davis [1], Yan-An Liu [2,3,4], Chelsea A. Corr [1], and Wei Gao [1,5]

[1]United States Department of Agriculture UV-B Monitoring and Research Program, Natural Resource Ecology Laboratory, Colorado State University, Fort Collins, CO 80523, USA
[2]Key Laboratory of Geographic Information Science (Ministry of Education), East China Normal University, Shanghai 200241, China
[3]School of Geographic Sciences, East China Normal University, Shanghai 200241, China
[4]ECNU-CSU Joint Research Institute for New Energy and the Environment, Shanghai 200062, China
[5]Department of Ecosystem Science and Sustainability, Colorado State University, Fort Collins, CO 80523, USA

*Correspondence to*: Maosi Chen (maosi.chen@colostate.edu), Zhibin Sun (zhibin.sun@colostate.edu), Yan-An Liu (yaliu@geo.ecnu.edu.cn)

**Abstract.** To recover the actual responsivity for Ultraviolet Multi-Filter Rotating Shadowband Radiometer (UV-MFRSR), the complex (e.g. unstable, noisy, and with gaps) time series of its in-situ calibration factors (Vo) need to be smoothed. Many smoothing techniques require accurate input uncertainty of the time series. A new method is proposed to estimate the dynamic input uncertainty by examining overall variation and subgroup means within a moving time window. Using this calculated dynamic input uncertainty within Gaussian Process regression (GP) provides the mean and uncertainty functions of the time

series. This proposed GP solution was first applied on a synthetic signal and showed significant smaller RMSEs than a Gaussian Process regression performed with constant values of input uncertainty and the mean function. GP was then applied to three UV-MFRSR Vo time series at three ground sites; The method appropriately accounted for variation in slopes, noises, and gaps at all sites. The validation results demonstrated that the agreement between aerosol optical depths (AODs) calculated using Vo determined by the GP mean function and AERONET AODs were consistently better than those calculated using Vo

from standard techniques (e.g. moving average). The improved accuracy of in-situ UVMRP Vo values suggests the GP solution is a robust technique for accurate analysis of complex time series and may be applicable to other fields.

## 1 Introduction

While many instruments generate relatively stable data time series over short time windows, dynamic uncertainty levels, variable sampling densities, and/or different lengths of gaps with missing data can complicate the analysis of long-term

datasets. For example, the five-year time series of a solar variability indicator (Mg II core to wing index) shows consistency on the order of days but increasing noise level and gaps are observed at the month-scale (Cebula et al., 1992). The time series



of the geopotential scale factor, a function of the geoidal potential, is also relatively stable on shorter time scales but demonstrates a slowly increasing long-term pattern (Burša et al., 1997). Additionally, the time series of a ratio (F factor) for calibrating a satellite radiometer suite (i.e. VIIRS) shows band-specific gap distributions and variable trends (Cardema et al.,

2012). As a result, these time series may not be described as a simple deterministic function of time due to possible noise and gaps.

Long term measurements of irradiance by Multi-Filter Rotating Shadowband Radiometers (MFRSRs) are also subject to errors imposed by the factors mentioned above. The MFRSR measures direct normal, diffuse horizontal, and total horizontal

irradiances at seven visible channels with a roughly 10 nm full half-maximum width (FHMW) (Harrison and Michalsky, 1994). The Ultraviolet (UV) version of MFRSR measures the same three irradiance components at seven UV channels (i.e. 300, 305, 311, 317, 325, 332, and 368 nm) with a 2 nm FHMW (Gao et al., 2010). Currently, the U.S. Department of Energy (DOE) Atmospheric Radiation Measurement (ARM) Climate Research Facility (Mather and Voyles, 2013), the NOAA Surface Radiation (SURFRAD) (Augustine et al., 2005) and the U.S. Department of Agriculture (USDA) UV-B Monitoring and

Research Program (UVMRP) (Gao et al., 2010) maintain their own MFRSR and/or UV-MFRSR at multiple sites across the U.S. To capture immediate instrument responsivity variation, the UVMRP performs in-situ calibrations using the Langley method (Slusser et al., 2000;Harrison and Michalsky, 1994) or derived approaches [e.g. (Chen et al., 2013;Chen et al., 2016;Chen et al., 2015)] on (UV-)MFRSR direct beam measurements on days with extended clear-sky periods (Gao et al., 2010).


Many factors contribute to the error or uncertainty of the Langley method including variations in aerosol and/or other atmospheric constituents over the course of the calibration period (Augustine et al., 2003;Chen et al., 2015;Zhang et al., 2016), the presence of thin cirrus (Shaw, 1976), as well as instrument errors (e.g. instrument tilt and misalignment, incorrect night-time offset and angular corrections) (Alexandrov et al., 2007). Thus, the sequence of original UVMRP (UV-)MFRSR in-situ

calibration factors exhibits certain levels of noise. Among these uncertainties, variable AOD is considered the major contributor to the variability of the Langley calibration factors obtained in typical atmospheric conditions over the continental United States (Alexandrov et al., 2008), even with careful cloud screening [e.g. (Chen et al., 2014;Alexandrov et al., 2004)]. In addition, extended cloudy periods and low solar zenith angles during winter months further reduce the sequence quality, which appear as large time gaps in the datasets. Since the in-situ calibration factor represents the instrument's responsivity

which is assumed to be relatively stable, it has been suggested that one applies some smoothing methods (e.g. averaging or fitting a smooth curve) on the daily calibration time series (Alexandrov et al., 2008) to reduce the issue. Currently, UVMRP implements an outlier detection and moving smoothing technique to overcome these issues. However, the process involves manual interaction, performs unreliably during sparse and gapped periods, and lacks the uncertainty estimation.




Analyses of complex long-term time series, such as those of (UV-)MFRSR Vo values, must consider: (i) the underlying continuous trend (i.e. the mean function) and the corresponding trend uncertainty and (ii) the (dynamic) input uncertainty. For problem (i), there is a variety of available approaches, such as local polynomial regression, smoothing splines, and Gaussian Process regression (Proietti, 2011). Local polynomial regression (LPR) constructs a polynomial within each local time window, and fits its coefficients by locally weighted least squares. LPR's computational complexity is low, and it can eliminate some of the randomness in the data (Hyndman, 2011). However, LPR may have difficulty on the cases with varying sampling densities or gaps. In addition, LPR does not allow estimating the trend near the ends of the time series and cannot be used for forecasting (Hyndman, 2011). A spline is a piecewise polynomial function with continuous derivatives (Proietti, 2011), and smoothing splines estimate the underlying spline by minimizing the distance between the spline and the observations while penalizing the roughness of the spline (Wahba, 2011). For example, a cubic spline fit was used to fill the large gaps in the Mg II index time series (Viereck et al., 2004). Both LPR and smoothing splines are unable to utilize the information about the input uncertainties or to estimate the uncertainty associated with the trend. Unlike the two methods above, Gaussian Process does not restrict the class of the underlying functions because it is not a parametric model (Rasmussen and Williams, 2006). Instead, it gives a priori probability to every possible function based on the desired function characteristics such as smoothness (Rasmussen and Williams, 2006). Gaussian Process regression assumes both the observations and the underlying function are from one joint (prior) Gaussian distribution, and derives the underlying function distribution by conditioning the joint (prior) distribution on the observations (Rasmussen and Williams, 2006). The method takes the observational error into consideration and naturally gives the uncertainty of the underlying function, making itself an appropriate tool for problem (i). Gaussian Process regression has been widely used in many fields [e.g. forecasting of mortality rates (Wu and Wang, 2018), prediction of spatial-temporal violent events (Kupilik and Witmer, 2018), and modelling received signal strength for wireless local area network location fingerprinting (Richter and Toledano-Ayala, 2015)].

For problem (ii), the input error statistics (e.g. input uncertainty) is often assumed to be known or roughly estimated in advance; In practice, a typical approach may use some predetermined constant (e.g. the nominal uncertainty of an instrument, or the standard deviation of its observation) to estimate input uncertainty for the entire dataset. However, this kind of approach omits the information of the possible time-varying observation error, leading to over- or under- estimation of the input uncertainty at a given (temporal) location (Chandorkar et al., 2017). A sophisticated approach may treat the dynamic input uncertainty as additional parameters and solve them together with other model parameters through optimization under the Bayesian framework (Kavetski et al., 2006a, b). However, this method requires the specification of valid error/uncertainty models, which are normally poorly understood in practice (Kavetski et al., 2006a, b).



In this study, we developed and validated a generic solution that combines Gaussian Process regression with a new dynamic
input uncertainty estimation method, to determine the underlying continuous trend and the corresponding uncertainty for the
given time series. In section 2, we briefly summarize the basics of the Gaussian Process regression and develop the dynamic
input uncertainty estimation method. We also describe a complex (noisy, gapped, etc.) synthetic time series and real UV-
MFRSR in-situ calibration factor time series used in the analysis. In section 3, we present and discuss the performance of the
Gaussian Process method on the test data, in comparison with the UVMRP current operational method and a moving average
technique. Validation of the calibration factors determined with the Gaussian Process method via the comparison of AODs
calculated with these factors and those reported by the AErosol RObotic NETwork (AERONET)(Holben et al., 1998) is also
discussed in section 3.

## 2 Materials and Methods

### 2.1 Gaussian Process regression (GP)

#### 2.1.1 Main Procedure

A Gaussian Process is a technique used in the analysis of a finite number of random variables with a joint Gaussian distribution
(Rasmussen and Williams, 2006). The following introduces briefly the theory of GP regression. An observed dataset,
$\mathcal{D}_{obs} = (\mathbf{X}, \boldsymbol{y}) = \left\{ (\boldsymbol{x}_i, y_i) \mid i = 1, \ldots, N \right\}$, has $N$ pairs of inputs ( $\mathbf{X} = \{\boldsymbol{x}_i\} \in \mathbf{R}^{N \times D}$ ) and corresponding observed values (
$\boldsymbol{y} = \{y_i\} \in \mathbf{R}^N$ ), where $D$ is the length of input vector $\boldsymbol{x}_i$. $\boldsymbol{y}$ is the combination of a function of $\mathbf{X}$ and noises $\boldsymbol{\varepsilon}$: $\boldsymbol{y} = \mathbf{f}(\mathbf{X}) + \boldsymbol{\varepsilon}$, where
$\boldsymbol{\varepsilon}$ follows an independent distributed Gaussian distribution $\boldsymbol{\varepsilon} \sim \mathcal{N}\left(\boldsymbol{0}, diag(\boldsymbol{\sigma}_\mathbf{y}^2)\right)$ and $\boldsymbol{\sigma}_\mathbf{y} \in \mathbf{R}^N$ is the given or estimated
uncertainty (standard deviation) on the $N$ observations. In practice, $\boldsymbol{\sigma}_\mathbf{y}$ is not always known in advance. The section below,
"Proposed Dynamic Input Uncertainty Estimation", provides an empirical approach to estimating $\boldsymbol{\sigma}_\mathbf{y}$. It is assumed that the
test dataset $\mathcal{D}_* = (\mathbf{X}_*, \boldsymbol{f}_*) = \left\{ (\boldsymbol{x}_{*i}, f_{*i}) \mid i = 1, \ldots, N_* \right\}$ and the observed dataset ( $\mathcal{D}_{obs}$ ) have the joint Gaussian distribution but the
test function values ( $\boldsymbol{f}_*$ ) are unknown:

$$\begin{bmatrix} \boldsymbol{y} \\ \boldsymbol{f}_* \end{bmatrix} \sim \mathcal{N}\left( \boldsymbol{0}, \begin{bmatrix} \mathbf{K}_{\mathbf{XX}} + \sigma_\mathbf{y}^2 \mathbf{I} & \mathbf{K}_{\mathbf{XX}_*} \\ \mathbf{K}_{\mathbf{X}_*\mathbf{X}} & \mathbf{K}_{\mathbf{X}_*\mathbf{X}_*} \end{bmatrix} \right), \tag{1}$$

where, $\mathbf{I}$ is the identity matrix, $\mathbf{K}_{\mathbf{X}_*\mathbf{X}} \in \mathbf{R}^{N_* \times N}$ denotes the covariance matrix between observed ( $\mathbf{X}_*$ ) and test inputs ($\mathbf{X}$), and
similarly for the other three terms $\mathbf{K}_{\mathbf{XX}} \in \mathbf{R}^{N \times N}$, $\mathbf{K}_{\mathbf{XX}_*} \in \mathbf{R}^{N \times N_*}$, and $\mathbf{K}_{\mathbf{X}_*\mathbf{X}} \in \mathbf{R}^{N_* \times N}$. Each element of these covariance
matrices is determined by a kernel function $K(z_1, z_2)$, which maps any pair of inputs ( $z_1, z_2 \in \mathbf{R}^D$ ) into $\mathbf{R}$. There are a wide





variety of kernel functions such as the radial basis function (RBF) and the rational quadratic (RQ) kernel (Rasmussen and
Williams, 2006). For example, The RQ kernel is defined by the following equation with length scale ($l$) and alpha ($\alpha$) as its
two parameters (Rasmussen and Williams, 2006):

$$k_{RQ}(r) = \left(1 + \frac{r^2}{2\alpha l^2}\right)^{-\alpha}, r = \|z_1 - z_2\|. \tag{2}$$

In practice, users need to use prior knowledge or techniques such as autocorrelation to choose the best kernel function to
represent the correlation among input data. The hyperparameters ($\boldsymbol{\theta}$) of the chosen kernel function are then optimized by
maximizing the log transformed marginal likelihood (Rasmussen and Williams, 2006):

$$\log p(\boldsymbol{y} \mid \mathbf{X}, \boldsymbol{\theta}) = -\frac{1}{2}\boldsymbol{y}^T \left[\mathbf{K}_{\mathbf{XX}}(\boldsymbol{\theta}) + \sigma_{\mathbf{y}}^2\mathbf{I}\right]^{-1}\boldsymbol{y} - \frac{1}{2}\log\left|\mathbf{K}_{\mathbf{XX}}(\boldsymbol{\theta}) + \sigma_{\mathbf{y}}^2\mathbf{I}\right| - \frac{N}{2}\log 2\pi . \tag{3}$$

To simplify the calculation, the mean of $\boldsymbol{y}$ has been subtracted from both the actual observed values and the test function
values. Therefore, the joint distribution has a mean equal to zero.


Based on the (optimized) joint distribution [Eq. (1)], the theorem that derives the conditional distribution from the joint
Gaussian distribution (Eaton, 1983), and the inversion equations of a partitioned matrix (Press, 1992), the Gaussian Process
regression predicts $\boldsymbol{f}_*$ from given $\mathbf{X}$, $\boldsymbol{y}$, and $\mathbf{X}_*$ (Rasmussen and Williams, 2006):

$$\{\boldsymbol{f}_* \mid \mathbf{X}, \boldsymbol{y}, \mathbf{X}_*\} \sim \mathcal{N}\left(\bar{\boldsymbol{f}}_*, \mathrm{cov}(\boldsymbol{f}_*)\right), \tag{4}$$

where,

$$\bar{\boldsymbol{f}}_* = \mathbf{K}_{\mathbf{X}_*\mathbf{X}}\left[\mathbf{K}_{\mathbf{XX}} + \sigma_{\mathbf{y}}^2\mathbf{I}\right]^{-1}\boldsymbol{y} , \tag{5}$$

$$\mathrm{cov}(\boldsymbol{f}_*) = \mathbf{K}_{\mathbf{X}_*\mathbf{X}_*} - \mathbf{K}_{\mathbf{X}_*\mathbf{X}}\left[\mathbf{K}_{\mathbf{XX}} + \sigma_{\mathbf{y}}^2\mathbf{I}\right]^{-1}\mathbf{K}_{\mathbf{XX}_*} . \tag{6}$$

The GP predicted sample standard deviations [i.e. the square root of the diagonal elements in $\mathrm{cov}(\boldsymbol{f}_*)$ ] can be converted to the
predicted confidence intervals. For example, the predicted 0.99999 confidence intervals used in this study are obtained by
multiplying a constant (i.e. 4.42) with predicted sample standard deviation. Points outside the predicted confidence intervals
may be considered as outliers and can be excluded iteratively until all points are within the confidence intervals or the average
ratio between GP predicted means and standard deviations are less than a threshold (e.g. the threshold is 0.01 in this study).

### 2.1.2 Proposed Dynamic Input Uncertainty Estimation

As mentioned before, the statistical properties of the noise $\boldsymbol{\varepsilon}$ of the observed time series $\boldsymbol{y}$ might be unknown. Even if assuming
$\boldsymbol{\varepsilon} \sim \mathcal{N}\left(\boldsymbol{0}, diag(\sigma_{\mathbf{y}}^2)\right)$ in practice, $\sigma_{\mathbf{y}}$ is not always a constant and could vary in time. Therefore, we propose to estimate $\sigma_{\mathbf{y}}$ with
a moving window approach. Within each moving window ($W$), the input uncertainty (denoted as $s_i$) is assumed to be relatively
stable and can be estimated using all points in the window ($W$). Note that $s_i$ is not equivalent to the standard deviation of all





points within the period ($s_W$), unless the mean function of the time series is invariant. We derive the relationship between $s_i$ and $s_W$ (see Appendix A for the detailed derivation) to estimate $s_i$:

$$s_W^2 = \frac{N-J}{N-1}s_i^2 + \frac{1}{N-1}\sum_{j=1}^{J} N_j\left(\mu_j - \mu_W\right)^2 , \tag{7}$$

where, all points within $W$ are clustered into $J$ subgroups based on their similarity in both time and value; $N_j$ is the number of points in each subgroup $j$; $N = \sum_{j=1}^{J} N_j$ is the number of all points within $W$; $\mu_j$ is the mean of subgroup $j$, which can vary among subgroups; $s_i$ is the estimated uncertainty of each point within $W$, acting as the sample standard deviation across all subgroups; $\mu_W$ and $s_W$ are the mean and sample standard deviation of all points within $W$. The classic K-Means algorithm was used for the clustering process. To increase the reliability to estimate statistics (mean or sample standard deviation), small subgroups are merged with adjacent ones to ensure each subgroup has more than required minimum points. The numbers of initial subgroups and the required minimum points depend on the prior knowledge on the variability and availability of the data. Sensitivity studies (not shown) indicate that 5 initial subgroups per moving window and 3 required minimum points per subgroup worked well for our applications. The dynamic input uncertainty estimation process is applied on every data point in a sequence. The squares of the estimated input standard deviations [i.e. $s_i^2$ in Eq. (7)] are stored on the respective diagnostic positions in $\sigma_y$.

The flowchart of the proposed Dynamic Input Uncertainty Estimation method and the complete GP procedure of estimating the mean and confidence interval functions of a given time series is presented in Figure 1.





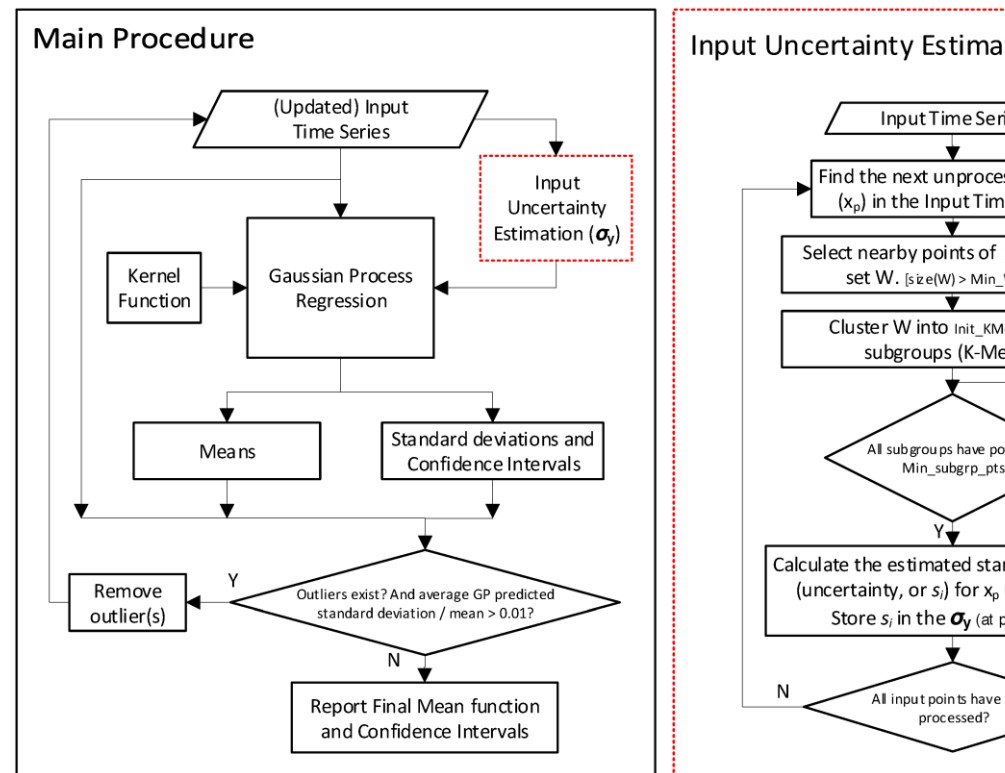

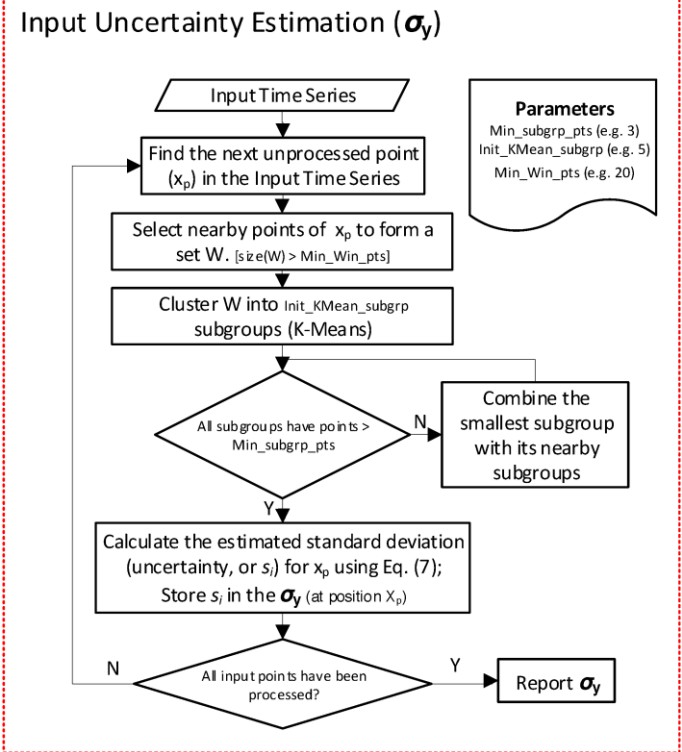

**Figure 1. Main procedure for deriving the mean and confidence interval functions using Gaussian Process Regression (left black box) and detailed procedure of the proposed Input Uncertainty Estimation method (right red box).**

**2.2 Moving Average (MA)**

Moving Average (MA) is a simple smoothing technique. To assess the performance of the GP regression with other methods, this study implements MA for one-dimensional case as follows. For a given $x_{*i}$, we first choose its nearby observations $\left\{ (x, y) \middle\| \lvert x - x_{*i} \rvert \leq win\_size, (x, y) \in \mathcal{D}_{obs} \right\}$ within the given window *win_size* and then calculate the mean y value of the subset as the smoothed observation at $x_{*i}$. The process is repeated for all possible x in $\mathcal{D}_{*}$.

**2.3 UVMRP operational algorithm (OPER)**

UVMRP operational algorithm (OPER) was specially designed for smoothing its in-situ calibration factor sequences (http://uvb.nrel.colostate.edu/UVB/dataProcessingInfo/VnaughtsDataProcessing.jsf). OPER is included as an additional source for methods comparison. The algorithm has three steps. In the first step, a 12-count running mean and the corresponding standard deviation are maintained to detect outliers (i.e. points outside half of the running mean or two standard deviation). During the process, if three consecutive points are determined to be outliers, visual examination is performed to determine if

a permanent change in the instrument responsivity has occurred. If such a change is confirmed, calculation of a new running



mean begins on the three points. In the second step, a moving linear regression is used to smooth the values at the center of each moving window. The moving window size is ±3 months. If visual examination finds significant value changes on a date of interest (the center of a moving window), the regression is not performed on that date. In the final step, the regression results from step two are used as input into a weighted means algorithm to generate continuous smooth in-situ calibration factors. The
inverse of year fraction between the current date of interest and the date of each participating point is used to calculate the weights. The weighting window is also ±3 months from the date of interest.

## 2.4 Validation method for 368-nm in-situ calibration factors

Since there are no other ground radiometers that measure the irradiance at the exact 368 nm channel as the UVMRP UV-MFRSR does, the estimated mean normalized Vo (Vo_norm) values from the Gaussian Process regression and the other two
comparison methods (i.e. MA and OPER) are validated indirectly in terms of aerosol optical depth (AOD) against those obtained at the collocated AERONET sites. AERONET sunphotometers are routinely calibrated with the uncertainty of AOD around 0.002 to 0.005 in the visible and up to 0.01 in the UV region (Eck et al., 1999;Holben et al., 2001) and are therefore considered a reliable source for AOD intercomparison and radiometer validation [e.g. (Alexandrov et al., 2002, 2008;Augustine et al., 2003;Krotkov et al., 2005a;Krotkov et al., 2005b;Kassianov et al., 2007;Tang et al., 2013;Yin et al.,
2015;Zhang et al., 2016)]. Augustine et al. (2003) compared SURFRAD MFRSR AODs at the Table Mountain station in Colorado with UVMRP MFRSR AODs at the Pawnee station (85 km northeast of Table Mountain) and with National Renewable Energy Laboratory (NREL) sun-photometer-derived AODs at Golden station (50 km to the south). The AOD difference on the test cases showed a magnitude of 0.1 to 0.2 and was variable over time even for the same comparison site. Krotkov et al. (2005a);Krotkov et al. (2005b) validated the UVMRP UV-MFRSR AODs with the interpolated AERONET
AODs at 368 nm at the National Aeronautics and Space Administration/Goddard Space Flight Center (NASA/GSFC) site in Greenbelt, Maryland. They found that the UV-MFRSR AODs at 368-nm channel on cloud-free days had a daily RMSE less than 0.01 when calibrated using AERONET measurements and increased to approximately 0.02-0.05 (depending on the season) when calibrated using standard Langley method (Harrison and Michalsky, 1994;Slusser et al., 2000). Alexandrov et al. (2002) developed an comprehensive calibration method for the VIS-MFRSR and validated the calibration at the four
channels (i.e. 440, 500, 670, and 870 nm) by comparing the derived AOD values with interpolated AERONET values at the ARM Cloud and Radiation Testbed (CART) site. The results showed small AOD difference (i.e. <0.005) at 440, 500, and 870 nm channels for a variety of atmospheric conditions with AODs ranging from 0.03 to 0.4 (at 500-nm). Alexandrov et al. (2008) considered optical depth of $NO_2$ and Ozone during the MFRSR AOD calculation, although they were small enough to be ignored (i.e. 0.008 $NO_2$ optical depth at 415 nm and 0.005 ozone optical depth at 615 nm) at their test location at the ARM
Southern Great Plains (SGP) site. The long-term intercomparison showed a good agreement (i.e. difference between them <0.01) between the MFRSR and AERONET AODs at 440, 675, and 870 nm channels. Kassianov et al. (2007) validated the MFRSR-retrieved optical properties and reported small RMSE values (i.e. 0.0043-0.0075) among MFRSR, AERONET, and


Normal incidence multifilter radiometer (NIMFR) derived AODs at 500 and 870 nm channels during the ARM Program's Aerosol Intensive Operational Period (IOP) in 2003.


In this study, for the UV-MFRSR at 368 nm channel, aerosol optical depth ($AOD_{368nm,UVMRP}$) is calculated by subtracting Rayleigh optical depth ($RLOD_{368nm,UVMRP}$) from total optical depth ($TOD_{368nm,UVMRP}$) under cloud-free conditions. The absorption of ozone, $NO_2$, and other trace gases are very small at the 368 nm channel (e.g. $NO_2$ optical depth is around 0.002 to 0.003 at AERONET Cart_Site), so they are ignored during the calculation of $AOD_{368nm,UVMRP}$:

$$AOD_{368nm,UVMRP} \approx TOD_{368nm,UVMRP} - RLOD_{368nm,UVMRP} . \qquad (8)$$

$TOD$ is calculated using Beer's Law (e.g. (Slusser et al., 2000)), where the actual calibration factor at top of atmosphere (Vo_raw) is restored from GP estimated mean Vo_norm. The cosine corrected voltage and airmass are obtained from the UVMRP webpage (https://uvb.nrel.colostate.edu/UVB/da_queryCosCorrected.jsf). $RLOD$ is calculated by following the equations in Bodhaine et al. (1999). The site latitude and height for $RLOD$ calculation are from the UVMRP webpage (https://uvb.nrel.colostate.edu/UVB/uvb-siteinfo.jsf), and the instantaneous site-level surface pressure for $RLOD$ calculation is obtained from the collocated AERONET sites (https://aeronet.gsfc.nasa.gov/cgi-bin/webtool_opera_v2_new).

To obtain reliable AOD values, UV-MFRSR measurements with quality concerns or cloud contamination are excluded in the following comparison. More specifically, (1) any measurements with UVMRP-provided quality control flag(s) relevant to the data quality of the direct beam at 368 nm channel are excluded; (2) data with small (direct beam) measurements at 368 nm are also excluded because they are more sensitive to noise or errors introduced during various calibration steps; and (3) a simple variation check is performed to reduce the potential of mixing cloud and aerosol optical depth. If the ratio between the standard deviation of TODs and the mean TOD value in the 15-minute time window exceeds 0.05, they are excluded from further analyses.

AERONET (v2.0) provides AOD at 340 and 380 nm channels. These values are interpolated to the effective wavelength of the UV-MFRSR 368 nm channel for comparison using the Ångström exponent as follows. Note that in the log transformed coordinate system [i.e. log(AOD) vs. log(wavelength)], log(AOD) is generally linear between 340 and 380 nm (Krotkov et al., 2005a). First, the AERONET AOD spectrum between the two wavelengths is derived by linear interpolation of AERONET AODs at 340 and 380 nm in the log transformed coordinate system. Next, since the UV-MFRSR AOD at 368 nm is a bandpass value over a narrow band (i.e 2 nm FHMW), the equivalent AERONET AOD at that channel is derived by

$$AOD_{368nm,AERONET} = \frac{\int_{340nm}^{380nm} AOD_\lambda F_\lambda S_\lambda d\lambda}{\int_{340nm}^{380nm} F_\lambda S_\lambda d\lambda} , \qquad (9)$$

where $AOD_\lambda$ is the interpolated AERONET AOD spectrum; $F_\lambda$ is the spectral response function of the UV-MFRSR at 368 nm channel (http://uvb.nrel.colostate.edu/UVB/da_queryFilterFunctions.jsf); $S_\lambda$ is the solar irradiance spectrum at top of





atmosphere (Chance and Kurucz, 2010); and the wavelength interval for the integral is 0.05 nm. Note that negative AERONET

AOD measurements are excluded from the validation because of using log transform.

## 2.5 Datasets

### 2.5.1 Synthetic Case

We generate a synthetic time series that is composed of six segments with a varying base function and noise levels [Figure 2

(a)]. The base function [Eq. (10)], including linear, quadratic, and cubic functions, simulates a wide variety of functions for

which the proposed technique is applicable. The noise levels are the same within each segment but different across segments.

The noise at segment $i$ is sampled from a fixed normal distribution $\mathcal{N}\left(0,\sigma_i^2\right)$, where $\sigma_i$ is equal to 4, 8, 6, 15, 7, and 3 from

left to right segments, respectively. Each segment originally contains 200 points. Their $x$ coordinates are sampled randomly

from six uniform distributions within their domains. Points with $x$ coordinates in the three designated windows (i.e. [64.2,

69.2], [80.8, 85.8], and [122.5, 127.5]) are removed to simulate data gaps in reality.

$$y = \begin{cases} 1.5x-30, & 0 \le x < 50 \\ -1.2(x-50)+45, & 50 \le x < 100 \\ -0.02(x-100)^2+2.3(x-100)-15, & 100 \le x < 150 \\ -0.02(x-150)^2-0.5(x-150)+50, & 150 \le x < 200 \\ 0.0004(x-200)^3+0.012(x-200)^2+0.4(x-200)-25, & 200 \le x < 250 \\ 0.002(x-250)^3-0.1(x-250)^2-2.5(x-250)+75, & 250 \le x < 300 \end{cases} \qquad (10)$$

### 2.5.2 Application Cases: In-situ calibration factors

In this study, the in-situ calibration factors of UVMRP UV-MFRSRs are used as application cases to test the performance of

the three smoothing methods (i.e. GP, MA, and OPER). These UV-MFRSR in-situ calibration factors over several months or

years are obtained through the Langley method on clear days. Their varying uncertainties are mainly attributed to two aspects.

One is the optical stability of atmospheric constituents (e.g., the aerosol, ozone, and thin clouds) when the in-situ calibration

factor is derived (Chen et al., 2015), and the other is the aging status of the radiometer. UVMRP publish its in-situ calibration

factors on their website (http://uvb.nrel.colostate.edu/UVB/da_queryVoIntercepts.jsf). To reduce the chances of abrupt

changes in the sequences, the data associated with the same instrument (i.e. UV-MFRSR) at the same UVMRP site (denoted

as a deployment period) are processed together. Three UVMRP sites with collocated AEROENT sites (for validation) were

selected (Table 1). The in-situ calibration factors at these UVMRP sites represent time series with contrasting densities,

noisiness, and slopes (Table 1). Appendix B uses the Oklahoma site (OK02) to show that the UV-MFRSR 368-nm in-situ

calibration factors obey normal distribution.

**Table 1. The three UVMRP 368-nm UV-MFRSR in-situ calibration factor time series for test.**



| UVMRP Site Name | UVMRP Site Location | Collocated AERONET Site | Deployment Start and End Dates | Figure (original time series) | Time Series Characteristics |
|---|---|---|---|---|---|
| HI02 | 19.54° N, 155.58° W, 3409 m | Mauna_Loa | 17 September 2015 to 1 July 2018 | Figure 3(a1) | dense, low noise, variable slope |
| IL02 | 40.05° N, 88.37° W, 213 m | BONDVILLE | 21 March 2017 to 29 May 2018 | Figure 3(b1) | sparse, high noise, sharper slope |
| OK02 | 36.60° N, 97.49° W, 317 m | Cart_Site | 17 January 2007 to 11 June, 2011 | Figure 3(c1) | medium density, medium noise, variable slope |



# 3 Results and Discussion

## 3.1 Synthetic Case

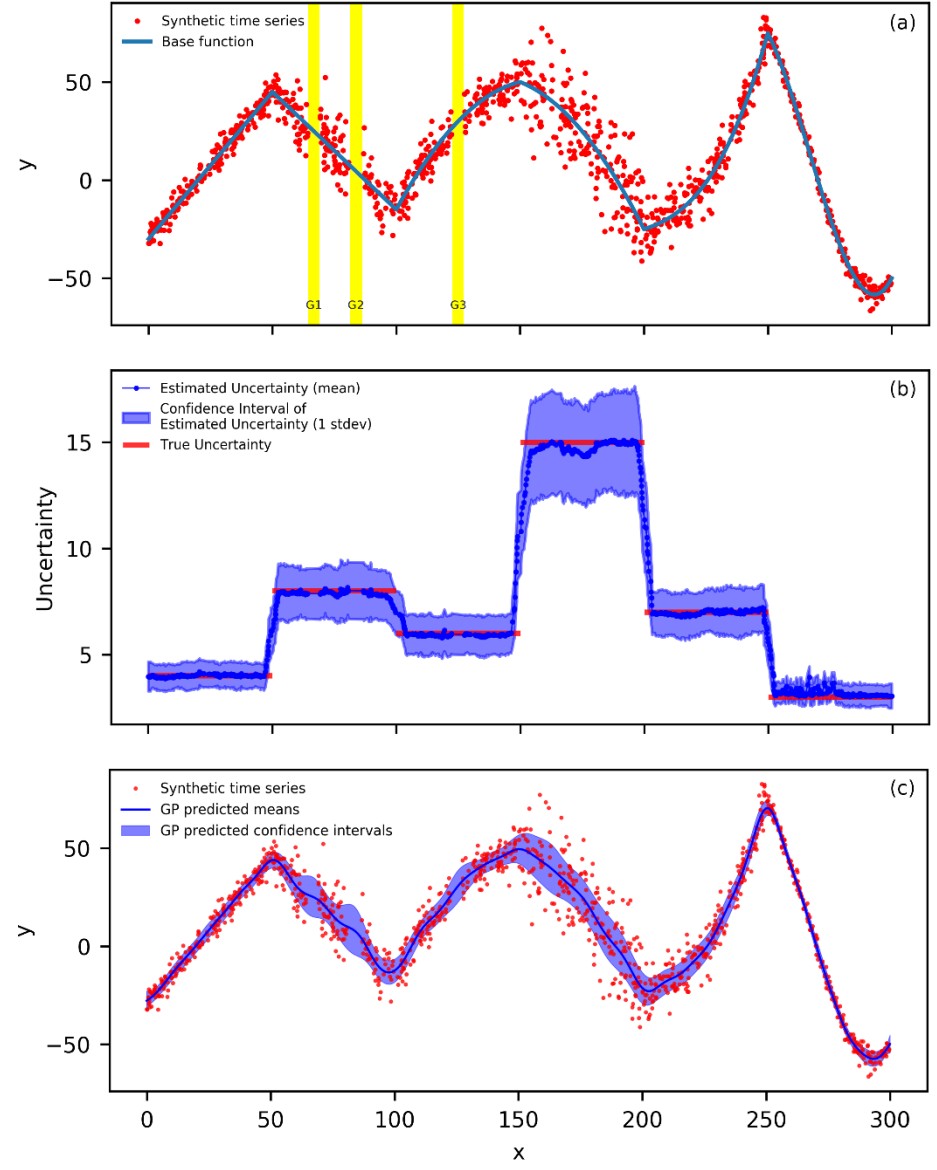

**Figure 2. (a) The synthetic time series based on Eq. (10) (the blue line) with varying noise levels. There are originally 200 samples within every 50-wide interval (or segment) in the x coordinate, but points between [64.2, 69.2] (highlighted in yellow, G1), [80.8, 85.8] (highlighted in yellow, G2), and [122.5, 127.5] (highlighted in yellow, G3) are removed to simulate data gaps in practice. The final number of points in the sequence is 1140. (b) The means (dark blue circles) and confidence intervals (light blue area) of the estimated uncertainty for the 200 synthetic sequences [all sampled from the distribution of (a) but with different random noise]. The true uncertainty (red line segments) is also displayed. (c) The Gaussian Process regression results on the synthetic time series from (a). The dark blue line is the predicted mean function and the light blue area is the corresponding confidence intervals.**



### 3.1.1 Estimation of Input Uncertainty for Gaussian Process

The proposed "Dynamic Input Uncertainty Estimation" method is first applied to the synthetic case. To observe the statistical properties/characteristics of the estimated input uncertainty, this procedure was applied on 200 synthetic time series, each of which is generated by adding random noise into the base function [Eq. (10)] following the procedures discussed in section "2.5.1 Synthetic Case".

Figure 2(b) shows the means (dark blue circles) and confidence intervals (light blue area) of estimated uncertainty of the 200 estimated input uncertainty sequences. The mean of the estimated uncertainty is close to the true uncertainty (RMSE = 0.6321) for the entire synthetic case as demonstrated by a linear regression between estimated and true uncertainty with a slope close to one (i.e. 1.0332) and a high $R^2$ of 0.9759 (Table 2). Most true uncertainty (red line segments) is covered by the confidence intervals except for the areas near the ends of the six segments. In these areas, the method averaged the uncertainty from the adjacent segments and presented a smooth transition between segments. This small RMSE value suggests that using smaller subgroup size (e.g., 3~6 points) does not significantly influence the estimation of uncertainty [Figure 2(a)]. Therefore, smaller subgroups are preferred over larger ones as larger subgroups are more likely to have gap(s) with large variation, which tends to increase its estimated standard deviation [Eq. (7)].

To demonstrate the improvements in the GP resulting from the dynamic input uncertainty estimation, the GP is also run with three typical constant input uncertainties: overall standard deviation of the synthetic time series (30.95), minimum true uncertainty of the synthetic time series (2.00), and maximum true uncertainty of the synthetic time series (15.00). The results from all three constant input uncertainties are less accurate than the estimated input uncertainty generated by the proposed method (Table 2). The proposed method has significant smaller RMSE (i.e. 0.6321) compared with the three constant input uncertainties (i.e. 24.1152, 6.5226, and 8.7921, respectively). Similarly, the linear regression between the estimated and true uncertainties shows that the proposed method has the slope and the $R^2$ values both close to one (i.e. 1.0332 and 0.9759) while the three constant uncertainties shows no (linear) correlation with true uncertainties (i.e. the slope and $R^2$ values close to zero).

### 3.1.2 Estimation of Means and Confidence Interval and Its Validation

The kernel function in the Gaussian Process regression used in this study is the rational quadratic (RQ) kernel, with two parameters: length scale and alpha [Eq. (2)]. To use RQ with Gaussian Process regression, we need to provide the initial (estimated) values for these two parameters. First, we round the original data points [red points in Figure 2(a)] to the nearest 0.25 interval grids. Then, we calculate the autocorrelation on these rounded data points from lags of 0.25 to 22.25 (approximately equivalent to lags of 1 to 90 points). Next, we perform curve fitting on autocorrelation results and obtain 9.80 and 1.05 as initial length scale and alpha estimates, respectively. With these initial RQ parameters and the estimated inputs uncertainty (from the proposed method or using three representative constant input uncertainties), Gaussian Process regression





predicts the mean and uncertainty functions. Figure 2(c) shows the GP results for the proposed method: dark blue line for the mean function and the light blue area for the confidence intervals (4.42 times of the GP predicted uncertainty function).

In terms of the GP predicted mean function vs. the base function [Eq. (10)], the proposed input uncertainty estimation method shows a 12.0% to 15.7% improvement on RMSE over the three constant input uncertainties (i.e. 1.1785 vs. 1.3146, 1.3976, and 1.3146) (Table 2). Similarly, the slope of the linear regression between the two functions is closer to one for the proposed uncertainty estimation method (i.e. 1.0082) than the three constant uncertainties (i.e. 1.0228). In addition, the predicted mean function from the proposed method is close to the base function even near the gaps [G1, G2, and G3 in Figure 2(a)] [Figure

2(c)].Additionally, the proposed method's predicted uncertainty function (or confidence intervals) shows better agreement with the true uncertainty of the synthetic time series [Figure 2(c)] while the three constant input uncertainties' results show consistent over- or under-estimated pattern over the entire time series (figures not shown). It is noted that the predicted confidence intervals from the proposed method are wider near the three gaps [G1, G2, and G3 in Figure 2(a)] than nearby locations with similar uncertainty. This is anticipated because the constraint in the gaps are from distant points where the RQ

kernel gives low correlation.

**Table 2 Validation of the input uncertainty and mean of GP prediction using four input uncertainties: the input uncertainty estimated by the proposed method (Section 2.1.2), overall standard deviation of the synthetic time series (30.95), minimum true uncertainty of the synthetic time series (2.00), and maximum true uncertainty of the synthetic time series (15.00). RMSE stands for**

**root mean square error. LR stands for linear regression. $R^2$ stands for the coefficient of determination for linear regression. Note: †**
**$y_1$ represents the true input uncertainty of the synthetic time series, $x_1$ represents the estimated input uncertainties. ‡ $y_2$ represents the true values on the base function [Eq.(10)], $x_2$ represents the GP estimated mean values using the respective input uncertainty.**

| | | | Constant input uncertainty | | |
|---|---|---|---|---|---|
| | Metrics | Proposed input uncertainty estimation method | Overall standard deviation (30.95) | Minimum synthetic time series uncertainty (2.00) | Maximum synthetic time series uncertainty (15.00) |
| input uncertainty | RMSE | 0.6321 | 24.1152 | 6.5226 | 8.7921 |
| | LR† | $y_1=1.0332x_1-0.2277$ | $y_1=-0.1962x_1-0.2277$ | $y_1=0.0x_1+7.1632$ | $y_1=0.0x_1+7.1632$ |
| | $R^2$ | 0.9759 | 0.0 | 0.0 | 0.0 |
| mean of GP prediction | RMSE | 1.1785 | 1.3146 | 1.3976 | 1.3146 |
| | LR‡ | $y_2=1.0082x_2-0.3865$ | $y_2=1.0228x_2-0.5351$ | $y_2=1.0228x_2-0.5636$ | $y_2=1.0228x_2-0.5351$ |
| | $R^2$ | 0.9986 | 0.9986 | 0.9983 | 0.9986 |





### 3.2 In-Situ Calibration Factors Cases

#### 3.2.1 Applications

The same GP procedure is applied on three in-situ calibration factor (Vo_norm, sun-earth distance normalized) sequences from three UVMRP deployment periods (Figure 3) at three different UVMRP locations previously described in Table 1.The Hawaii site (HI02) sits at a clean, high altitude location, which means its atmospheric condition is more stable than other UVMRP sites and its Vo_norm has the lowest variation [Figure 3(a1)]. The Illinois site (IL02) is surrounded by croplands/rangelands with the closest city (Champaign) located 12 km northwest [Figure 3(b1)]. The Oklahoma site (OK02) is also surrounded by croplands/rangelands with the closest city (Oklahoma City) located about 96 km south [Figure 3(c1)]. Both wildfires and agricultural activities (e.g. cultivation and harvest) at IL02 and OK02 contribute to the relatively hazy and unstable atmosphere condition for Langley regression. As the result, Vo_norms at IL02 and OK02 have larger variation compared with HI02. The dynamic input uncertainty estimation results confirm that the uncertainty at HI02 [15–40, Figure 3(a2)] is also lower than the other two sites [100–300, Figure 3(b2), (c2)]. Generally, the proposed method gives lower uncertainty values for time windows with more clustered points (e.g. December 2008 and April 2010 at OK02 [Figure 3(c2)], and February 2017 at HI02 [Figure 3(a2)]). There are no obvious temporal patterns of uncertainty at any of the three sites.

Figure 3(a3), (b3), and (c3) show the estimated means (dark blue line) and confidence intervals (light blue area) after the initial pass through GP. The length scale parameter of the RQ kernel for the HI02, IL02, and OK02 sites are 6.091, 6.369, and 6.228 (days), respectively. Their corresponding alpha parameters of the RQ kernel function are all close to 1.0 (i.e. 0.948, 0.862, and 0.944, respectively). As expected, the confidence interval is narrower near time windows with more data points, and the confidence intervals are wider near gaps [Figure 3(b3)].

As depicted in Figure 1, the outlier removal and GP are repeated following the initial GP regression, giving the final GP results shown in Figure 3(a4), (b4), and (c4). After this final pass, the length scale parameter of the RQ kernel function for the HI02, IL02, and OK02 sites are 6.091, 11.149, and 6.907 (days), respectively. Compared with the first round, all length scale parameters increase as more outliers are removed (except for HI02). At HI02, the average ratio between GP means and standard deviations is lower than the threshold (i.e. 0.01) after the first round and the iteration stops. The corresponding alpha parameters of the RQ kernel function are still all close to 1.0 (i.e. 0.948, 1.010, and 1.110, respectively). Because of outlier removal, compared with the first-round results, GP generates smoother mean functions and narrower confidence intervals at the last round.

The other two methods (i.e. MA and OPER) are applied on the same in-situ calibration time series. They can provide mean functions but not confidence intervals. The MA (win_size=20) results [Figure 3(a5), (b5), and (c5)] are generally smoother than OPER [Figure 3 (a6), (b6), and (c6)] but both are more responsive to noisy points than GP. In addition, since OPER is scheduled to run once per month on active deployments, there may be some lags at the end of those deployments [e.g. Figure 3(a6)].

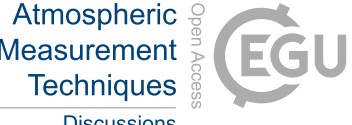



**Figure 3. The results of the three smoothing methods (i.e. GP – Gaussian Process, MA – Moving Average, OPER – UVMRP operational algorithm) on the three UVMRP in-situ calibration factor sequences: (a) HI02 (17 September 2015 to 1 July 2018), (b) IL02 (21 March 2017 to 29 May 2018), and (c) OK02 (17 January 2007 to 11 June, 2011). The first row (a1, b1, c1) displays the original in-situ calibration factor (Vo_norm) sequence. The second row (a2, b2, c2) shows the initial input uncertainty estimated for GP. The third row (a3, b3, c3) presents the predicted daily mean and confidence interval from the first iteration of GP. The fourth row (a4, b4, c4) shows the final results of GP after iterations. The fifth row (a5, b5, c5) shows the results of MA. The sixth row (a6, b6, c6) shows the results of OPER.**





## 3.2.2 Validation

**Figure 4. 368-nm AOD scatter plots between UVMRP (*y* axis) and AERONET (*x* axis). The UVMRP 368-nm AODs are calculated**
**from UV-MFRSR direct normal voltages using calibration factors estimated by the three methods (i.e. from top to bottom: GP, MA, OPER) at the three sites (i.e. from left to right: HI02, IL02, OK02). The AERONET 368-nm AODs are derived from collocated (i.e. Mauna_Loa, BONDVILLE, Cart_Site) AERONET AODs on the 340- and 380-nm channels. The linear regression line (solid, red) and the 1-by-1 line (dashed, black) are also plotted. "<*y-x*>" means the average difference between AERONET and UVMRP AOD at 368 nm channel. "stdev(*y-x*)" means the standard deviation of their difference.**





Following the procedures described in section 2.4, the UVMRP AODs at 368 nm channel generated by GP, MA, and OPER are validated against the corresponding AERONET AODs at the three collocated sites (i.e. HI02 – Mauna_Loa, IL02 – BONDVILLE, OK02 – Cart_Site). The scatter plots between these UVMRP and AEROENT AODs are displayed in Figure 4. The performance of all three methods at HI02 [Figure 4(a), (d), (g)] are similar. For example, the average bias "<y-x>" is approximately 0.0054 and standard deviation of the difference "stdev(y-x)" is approximately 0.0066. For IL02 [Figure 4(b),

(e), (h)] and OK02 [Figure 4(c), (f), (i)], GP shows superior agreements with AERONET than the other two methods. For example, at IL02, the absolute value of GP's average bias (0.0036) is about 3.3 to 2.5 times lower than that of MA (0.0119) and OPER (0.0091). Similarly, at OK02, the average bias for GP (0.0032) is much lower than those for MA (0.0119) and OPER (0.0087). The validation results for GP at OK02 are similar to the previous comparison results between AERONET and MFRSR AODs at 415 and 440 nm (Tang et al., 2013;Alexandrov et al., 2008).


**Table 3 Statistical metrics (average absolute difference, average absolute relative difference, and linear regression) on comparing 368-nm AOD between UVMRP ($AOD_{368,UVMRP}$) and AEROENT ($AOD_{368,AE}$). The UVMRP 368-nm AODs at the three sites (i.e. HI02, IL02, and OK02) are calculated using calibration factors estimated by the three methods (i.e. GP, MA, and OPER). The AERONET 368-nm AODs are derived from collocated (i.e. Mauna_Loa, BONDVILLE, and Cart_Site) AERONET AODs on the 340- and 380-**

**nm channels. LR stands for linear regression. $R^2$ stands for the coefficient of determination for linear regression. The "x" and "y" in "Linear regression equation" refer to $AOD_{368,AE}$ and $AOD_{368,UVMRP}$ of the respective methods.**

| Site | Metrics | Method | | |
|---|---|---|---|---|
| | | GP | MA | OPER |
| HI02 | $Avg(\|AOD_{368,UVMRP}-AOD_{368,AE}\|)$ | 0.0062 | 0.0065 | 0.0067 |
| | $Avg\left(\dfrac{\|AOD_{368,UVMRP}-AOD_{368,AE}\|}{AOD_{368,AE}}\right)$ | 0.5803 | 0.6078 | 0.6261 |
| | LR | y=1.0550x+0.0045 | y=1.0551x+0.0047 | y=1.0601x+0.0043 |
| | $R^2$ | 0.9000 | 0.8957 | 0.8812 |
| IL02 | $Avg(\|AOD_{368,UVMRP}-AOD_{368,AE}\|)$ | 0.0228 | 0.0291 | 0.0270 |
| | $Avg\left(\dfrac{\|AOD_{368,UVMRP}-AOD_{368,AE}\|}{AOD_{368,AE}}\right)$ | 0.1669 | 0.2087 | 0.1930 |
| | LR | y=0.9615x+0.0115 | y=0.9543x+0.0213 | y=0.9241x+0.0065 |
| | $R^2$ | 0.9514 | 0.9420 | 0.9332 |
| OK02 | $Avg(\|AOD_{368,UVMRP}-AOD_{368,AE}\|)$ | 0.0150 | 0.01785 | 0.01847 |
| | $Avg\left(\dfrac{\|AOD_{368,UVMRP}-AOD_{368,AE}\|}{AOD_{368,AE}}\right)$ | 0.1714 | 0.2067 | 0.1939 |
| | LR | y=1.0054x+0.0027 | y=1.0238x+0.0078 | y=1.0186x+0.0056 |
| | $R^2$ | 0.9749 | 0.9726 | 0.9554 |





Table 3 shows two additional statistical metrics for validation: "Avg($|AOD_{368,UVMRP}-AOD_{368,AE}|$)", a measure of absolute difference between the two quantities and "Avg($|AOD_{368,UVMRP}-AOD_{368,AE}|/AOD_{368,AE}$)" a measure of relative difference between the two quantities. For HI02, the GP Vo_norm values improves both the absolute (~4.5%) and relative (~7.5%) differences between $AOD_{368,UVMRP}$ and $AOD_{368,AE}$ when compared to MA and OPER AODs, respectively. Results from linear regressions (LR) performed between $AOD_{368,UVMRP}$ and $AOD_{368,AE}$ are also reported in Table 3. The LR results are similar between GP and MA, but GP has closer-to-one LR slope (1.0550) and higher $R^2$ (0.9000) than those of OPER (1.0601 and 0.8812) for HI02. For IL02, GP shows 21.6% smaller absolute difference and 20.0% smaller relative difference to AERONET than MA; GP shows 15.6% smaller absolute difference and 13.5% smaller relative difference to AERONET than OPER. Similarly, for OK02, GP shows 16.0% smaller absolute difference and 17.1% smaller relative difference to AERONET than MA; GP shows 18.8% smaller absolute difference and 11.6% smaller relative difference to AERONET than OPER.

Overall, the 368-nm AODs by GP shows higher correlation, closer-to-one slopes, and lower absolute and relative biases compared to AERONET AODs than MA and OPER at all three sites. The improvement of GP over MA and OPER at IL02 and OK02 are more significant than at HI02. The main reason may be that HI02 is the least polluted site among the three sites. Both of its maximum and mean 368-nm AOD values are low: 0.35 and 0.016, respectively. As a result, higher accuracy of Rayleigh and other optical depth components to discern small improvement on AOD for HI02. Since the AERONET's sun photometer is routinely calibrated, the agreement on AOD values suggests that the calibration factors mean function generated by GP are more accurate than MA and OPER.

In addition, Figure 5 shows the 368-nm AOD time series calculated using GP generated in-situ calibration factors at the three UVMRP sites. The blue solid line represents the AODs calculated using the GP means, and the green and red dotted lines represent the AODs calculated using the GP confidence intervals. It is seen that the AOD confidence intervals are approximately ±0.0095, ±0.0480, and ±0.0273 at HI02, IL02, and OK02, respectively. The corresponding AERONET AOD time series are also plotted (i.e. purple lines in Figure 5). The insets in Figure 5 show comparison details at HI02, IL02, and OK02. For most of the AOD time series, AERONET results are within the GP confidence intervals. The average absolute differences of daily AOD values between GP and AERONET are ~0.006 for HI02, ~0.024 for IL02, and ~0.014 for OK02. These values are close or within the AERONET AOD uncertainty level (i.e. 0.01), suggesting the high-quality of the potential UVMRP AOD product. In addition, unlike the obvious seasonal changes in AOD difference reported in the previous study at the NASA/GSFC site by Krotkov et al. (2005a), this study (Figure 5) shows no discernible seasonal pattern in the AOD differences at all three sites.



**Figure 5. Time series of UVMRP and AERONET 368-nm daily average AOD at HI02, IL02, and OK02 sites. The daily AOD mean values derived from the GP mean in-situ calibration factor ($V_o$) functions (blue) and the corresponding AERONET values (purple) are shown in blue solid lines. The corresponding lower and upper limits of AOD derived from the GP $V_o$ confidence intervals are shown in dotted red and green lines, respectively. The insets for HI02 (August 2016), IL02 (June 2017), and OK02 (July 2010) are also included in the respective subplots to show the comparison details.**





## 4 Conclusions


A new dynamic uncertainty estimation method for noisy time series is developed in this study. Combining this method with Gaussian Process regression, we provide a solution to estimate the underlying mean and uncertainty functions of time series with variable mean, noise, sampling density, and length of gaps. For the synthetic case with linear, quadratic, and cubic base functions, noise level varying from 2 to 15, and noticeable gaps, the proposed solution returns a mean function with the RMSE

of 1.1785 (linear regression $R^2$ of 0.9986), which is at least 12.0% lower than RMSEs associated with the three constant input uncertainties. Its estimated input uncertainties determined by this method are close to the true uncertainty levels except for the transitional region between segments. The solution also gives accurate mean values at the three gaps. The proposed GP solution as well as the other two comparison methods (i.e. MA and OPER) were then applied on three in-situ calibration factor time series of UV-MFRSR (368 nm) at three UVMRP sites. The GP solution handles the variation in slope, noise, sampling density,

and length of gap in the three cases as expected. Since irradiance at 368 nm is not measured by a collocated (and calibrated) radiometer, the performance of the three methods is validated against the collocated AERONET sites in terms of AOD. The results show that AODs calculated using GP-derived UV-MFRSR calibration factors (Vo_norm) have consistently better agreement with AERONET AODs than MA and OPER in terms of average absolute and relative differences, and linear regression $R^2$ values. These results suggest that the proposed GP solution is a robust method for time series analyses of data

with variable mean, noise, sampling density, and length of gap, and has potential for application across disciplines..

**Appendix A. The formulation between the overall standard deviation and the subgroup standard deviation**

Given a time series $\{x_i\}$, its total $N$ points are divided into $J$ groups $\{x_k^j\}$, and the number of points in group $j$ is $N_j$ ($j$=1,2,…,$J$; $k$=1,2,…, $N_j$). For data points in each group, their sample mean and standard deviation are $\mu_j$ and $s_j$. For the entire time series, its sample mean is $\mu = \dfrac{1}{N}\sum\limits_{i=1}^{N} x_i$, and the sample variance is


$$
\begin{aligned}
s^2 &= \frac{1}{N-1}\sum_{j=1}^{J}\sum_{k=1}^{N_j}(x_k^j - \mu)^2 \\
&= \frac{1}{N-1}\sum_{j=1}^{J}\sum_{k=1}^{N_j}[(x_k^j - \mu_j) + (\mu_j - \mu)]^2 \\
&= \frac{1}{N-1}\sum_{j=1}^{J}\sum_{k=1}^{N_j}(x_k^j - \mu_j)^2 + \frac{1}{N-1}\sum_{j=1}^{J}\sum_{k=1}^{N_j}(\mu_j - \mu)^2 + \frac{2}{N-1}\sum_{j=1}^{J}\sum_{k=1}^{N_j}(x_k^j - \mu_j)(\mu_j - \mu) \\
&= \frac{1}{N-1}\sum_{j=1}^{J}N_j s_j^2 + \frac{1}{N-1}\sum_{j=1}^{J}N_j(\mu_j - \mu)^2
\end{aligned}
$$
,





where the third term on the right-hand side is equal to zero because $\sum_{k=1}^{N_j}(x_k^j - \mu_j) = 0$ (i.e. $\mu_j = \frac{1}{N_j}\sum_{k=1}^{N_j}x_k^j$). If assume that the

sample standard deviation of each data point is invariant (i.e. $s_1 = s_2 = \cdots = s_J = \hat{s}$), then

$$s^2 = \frac{N-J}{N-1}\hat{s}^2 + \frac{1}{N-1}\sum_{j=1}^{J}N_j(\mu_j - \mu)^2 \quad .$$

**Appendix B. The distribution of the 368-nm in-situ calibration factors of UV-MFRSR.**

Since the true 368-nm in-situ calibration factors are not available, their distribution is derived using the AEROENT 368-nm

AOD distribution via Beer's Law (transformed Langley regression).

Beer's Law links the irradiance [or voltage ($V$)] at top of atmosphere with the one reaches ground at time t with the equation:

$V_t = V_o e^{-TOD_t \cdot m_t}$ , where $m_t$ is the airmass at time $t$ and $TOD_t$ is the corresponding total optical depth. For the 368-nm channel,

$AOD$ is the main contributor for the $TOD$ variation. Therefore, for a short time period, $TOD_t$ can be expressed as the sum of a

constant optical depth ($\bar{P}$) and variable residual aerosol optical depth ($\Delta AOD_t = AOD_t - \overline{AOD}$): $TOD_t = \bar{P} + \Delta AOD_t$ . To

derive an unbiased $V_o$, Langley regression (in the transformed lnV·m$^{-1}$ vs. m$^{-1}$ coordinate system), it requires the participating

measurements have a constant $TOD_t$ over the calibration period and ln$Vo$ is the slope of the regression. $\Delta AOD_t$ biases to the

regression slope with the component varying linearly with $m_t^{-1}$ (Chen et al., 2014). Therefore, we decompose $\Delta AOD_t$ as the

sum of a constant term ($\alpha$) and a $m_t^{-1}$ term ($\beta m_t^{-1}$), where $\alpha$ and $\beta$ are obtained from daily AERONET 368-nm AOD

measurements via linear regression. With the $TOD_t$ components expanded, the original Beer's Law equation is expressed as

$\ln V_t \cdot m_t^{-1} = -\left(\bar{P} + \alpha\right) + \left(\ln V_o - \beta\right)m_t^{-1}$ and the (transformed) Langley regression obtains the slope ($\ln V_o = \ln V_o - \beta$) via linear

regression. The disturbed distribution of $V_o$ is the same as the distribution of $\exp(\ln V_o - \beta)$. Assuming the true $V_o$ is 1500 mV

(a typical value at OK02) and using a long-term set of $\beta$ values from AERONET at Cart_Site (17 January 2007 to 11 June,

2011), a set of $V_o$ is obtained. Removing the tails on the distribution of $V_o$ (i.e. $V_o$ <1200 or $V_o$ >1800), the normal test of the

$V_o$ set (using the Python function scipy.stats.normaltest (D'Agostino and Pearson, 1973)) returns the p value of 0.4689, which

is greater than the threshold ($10^{-3}$), suggesting that the $V_o$ set comes from a normal distribution.

**Data availability**

The in-situ calibration factors (sun-earth distance normalized) used in this study were downloaded from the UVMRP website:
http://uvb.nrel.colostate.edu/UVB/da_queryVoIntercepts.jsf. The cosine corrected voltage and airmass were obtained from





https://uvb.nrel.colostate.edu/UVB/da_queryCosCorrected.jsf. The spectral response functions of the UV-MFRSRs were obtained from http://uvb.nrel.colostate.edu/UVB/da_queryFilterFunctions.jsf. The site latitudes and heights of the three UMVRP sites tested in this study were obtained from https://uvb.nrel.colostate.edu/UVB/uvb-siteinfo.jsf. The solar irradiance spectrum at top of atmosphere was obtained from Chance and Kurucz, 2010. The AEROENT (v2.0) data (i.e. aerosol optical depth and surface pressure) used in this study were downloaded from https://aeronet.gsfc.nasa.gov/cgi-
bin/webtool_opera_v2_new.

**Author Contributions**

Authors Maosi Chen and Zhibin Sun are equally significant contributors to the research. Methodology, M. Chen and Z. Sun; Software, M. Chen; Analysis, M. Chen, Z. Sun, and J.M. Davis; Validation, M. Chen, Z. Sun and Y.-A. Liu; Writing-Original Draft Preparation, M. Chen and Z. Sun; Writing-Review & Editing, M. Chen, Z. Sun, C. Corr, J. M. Davis, Y.-A. Liu and W.
Gao; Supervision, J.M. Davis and W. Gao; Project Administration, W. Gao; Funding Acquisition, W. Gao.

**Acknowledgments**

This work is supported by the US Department of Agriculture (USDA) UV-B Monitoring and Research Program, Colorado State University, under USDA National Institute of Food and Agriculture Grant 2016-34263-25763. We thank Rick Wagener (PI) and the team for the effort in establishing and maintaining the U.S. Southern Great Plains (SGP) Cloud and Radiation
Testbed (CART) Site. We thank Brent Holben (PI) and the team for the effort in establishing and maintaining the AERONET Mauna_Loa site. We thank Brent Holben, Christopher M.B. Lehmann, and their team in establishing and maintaining the AERONET BONDVILLE site.

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
