# Peer review of "Improving the Mean and Uncertainty of Ultraviolet Multi-Filter Rotating Shadowband Radiometer In-Situ Calibration Factors: Utilizing Gaussian Process Regression with a New Method to Estimate Dynamic Input Uncertainty"

_Atmospheric Measurement Techniques, 2018_

## Referee Comment (RC1) · Anonymous Referee #2 · 13 Dec 2018

Manuscript: AMT-2018-295

General Comments: The manuscript presents a method that utilizes the Gaussian Process Regression with improved input dynamic uncertainty for calculating in-situ calibration factors for UV-MFRSRs. There are known sources of uncertainties and assumptions to the Langley technique as the author's note in the paper. Reducing the uncertainties in the calibration factors for calculating AOD from these types of instruments is crucial, as it also helps improve the ability to retrieve other aerosol optical properties.

The manuscript overall is structured and outlined well for ease of reading, the language is fluent and precise, and there is in general proper credit and relevant references. The manuscript does a good job detailing the GP technique, the new improvements incorporating dynamic input uncertainty, and comparing to two other techniques currently in use including their current operational procedure (OPER), and the moving average (MA). Since the improvements are modest across the techniques at HI02, the authors have included three sites (HI01, OK02, IL02) that show the improvements in GP over the two other methods are consistent. I believe this paper has shown the robustness of the GP method, and provides useful descriptive comparisons between techniques that will be helpful for operators of this type of instrument and how to improve the calibration in-situ. The paper addresses relevant questions within the scope of AMT as this paper is an investigation and validation of an improved method for calibrating a remote sensing instrument widely used within the atmospheric community for aerosol optical properties.

Specific Comments: In the abstract, I suggest including a sentence with the improved calibration numbers between the three methods, moving average (MA), the current operational version (OPER), and the Gaussian Process (GP) with improved dynamic input uncertainty for at least one site (HI01, IL02, or OK02).

Pg 8, line 193: Just a note to correct the wording of this sentence (though the sentence refers to irradiance at 369-nm). The Physikalish-Meteorological Observatorium Davos, World Calibration Center has a Precision Filter Radiometer (PFR) that measures AOD at 368-nm. Using this type of instrument would avoid additional uncertainties in AOD caused by the interpolation between wavelengths when comparing the MFRSR with the AERONET CIMEL. At the sites used for the comparison in this paper, the site HI02 has a PFR but I do not know about the other two sites. This isn't essential for the analysis, nor conclusions of the paper, only suggest the sentence be modified.

For validation of the technique, the authors compare AOD at 368-nm from the UV-MFRSR indirectly to the AERONET CIMEL using information of AOD at two wavelengths (340 and 380 nm). Different types of measurement techniques have their own source of uncertainties as with the CIMEL and the addition of the few paragraphs on previous literature that highlights these differences is crucial to the understanding the improvements using the GP technique.

Pg. 7, section 2.2 on Moving Average. This doesn't describe the moving window size used in the analysis.

Technical corrections: Pg 18, line 395. AEROENT needs to be AERONET. Pg 420, line 422-423. Incomplete sentence.

---

## Referee Comment (RC2) · Anonymous Referee #1 · 17 Dec 2018

This is a very interesting paper and also very helpful for UVMFR and MFRSR users in order to try to understand and improve a very important aspect that is the instrument calibration and the uncertainty of the calculated calibration constants. I think there is some room for improvement on basic aspects but there are no issues that would suggest the rejection of this work from AMT.

Concerning the introduction

[Figure]

I am missing previous results of (UV-) MFRSR comparisons with other standard AOD measuring instruments. For example in the 2015 the Filter Radiometer comparison in Davos, Switzerland various of this instrument types have participated and the results have been discussed. There are also earlier studies of such comparisons.

In Kazadzis et al., 2018: There were 4 MFRSR instruments in this campaign. The results were summarized in the following paragraph: "The four MFR instruments showed good agreement for the medians compared to the PFR triad, however, they exhibit larger scatter than the sun-pointing instruments resulting in a lower precision. McArthur et al. (2003) had previously reported that the MFR-derived AOD does not quite meet the accuracy of the sun-pointing instruments under clean atmospheric conditions. MFR_DE showed an AOD overestimation in various instances that gave results that are outside the WMO defined AOD limits (Fig. 2d). This small overestimation of the MFR_DE instrument compared to the PFR Triad could be due to uncertainties introduced while correcting for their angular response, the calibration procedure, or incomplete blocking of the diffuser by the shadow-band. The MFRSRs that are part of the SURFRAD network (MFR_US2 and MFR_US3) give a median AOD at 500-nm that is in very good agreement with the PFR triad and as good or better than some of the other sun-pointing instruments, e.g., CIMEL and POM; these two slightly underestimate the AOD at 865 nm, but are within the WMO defined limits. Again, these two MFRs' medians are comparable to the better sun-pointing instruments, but give larger scatter. These two MFRs are representative of the SURFRAD network that follow network protocols for calibration and alignment and frequent characterizations of the spectral and angular responses (Augustine et al., 2003, Michalsky et al., 2001)."

Would be helpful some of the above aspects to be included in the introduction section or in section 2.4

Authors are mentioning: "There are no other instruments measuring at 368nm" The World Meteorological Organization (WMO) instigated the Global Atmosphere Watch (GAW) program in 1989. Based on a recommendation by GAW experts, the World

[Figure]

Optical Depth Research Calibration Center (WORCC) was established in 1996 at the PMOD/WRC in Switzerland. WORCC has since been advised by the GAW Scientific Advisory Group for Aerosols. The standard instrument consist of a precision Filter radiometer (PFR measuring at 368, 412, 500 and 862 nm. So actually the WMO reference instruments (triad) is measuring at 368nm.

So the argument of the non existence of instruments measuring at 368nm (thus the choice of the AOD based comparison) is not correct. However, as it is not possible to repeat this study with one of the PFR instruments, probably the only solution could be the AOD comparison with the collocated cimels as the authors have initiated. Nevertheless, a short comment on the above text could be included in the paper.

Line 224 : probably you should comment also that ozone is also ignored .

Line 235: I guess that the cloud flagging method is not evaluated here, as comparison with CIMEL data includes only data that CIMEL algorithm considers as cloud free.

Line 250 : It would be informative to explain why S(lamda) appears in equation 9. Since F(lamda) is ∼4nm the integrated range is ∼366-370 nm. There you mention that AOD is the "interpolated ADO spectrum" which I guess you mean the linear (?) interpolated using 340 and 380 AERONET AODs ? Then S(lamda) is used for normalization in this small 4 nm range? Is this so different that the actual interpolated value of AOD at 368 nm? And if S(lamda) is used, why not S(lamda) – Rayleigh optical depth ? Have in mind that spectral function FWHM of the CIMEL is larger than 2nm.

The authors chose to evaluate their method by comparing the retrieved (from their Vos data) AODs. Here are my comments on this section:

- The comparison of UVMFR with AERONET would be essential to follow criteria that are defined by WMO –CIMO in order to assess the results in detail. https://library.wmo.int/pmb_ged/wmo-td_1287.pdf There (page 8) such conditions and formulas are defined. For example the U95 criterion where a number (here the lower

limit is 95%) of measurements have to be in the range of $\pm 0.005 + 0.010$/m Where 0.005 accounts for instrument related uncertainties and 0.01/m for calibration related uncertainties (calibration uncertainty better than 1%).

Changing the analysis figures with the use of this criterion authors can:

a. better show the agreement and the improvements with their methods by showing the percentage of data within these limits for each case.

b. having in mind that calibration related uncertainties will be inherited in AOD retrievals as a function of air mass, the figures (a) including aod differences vs air mass can point out on Vo related issues. Still slopes and cor. Coefficients can be reported in the form of a table.

The AOD retrieval and the differences among two instruments are a consequence not only an uncertainty on the instrument calibrations but also other factors.

Here is a list:

- The calculation of Rayleigh optical depth from both instruments including the pressure measurement. Are the two instruments ( UVMFR and CIMEL) use the same formulas ?

- The calculation of Rayleigh and aerosol air mass factors

- The potential differences in the field of view of the instrument

- CIMEL includes NO2 and Ozone optical depths

- The wavelength interpolation from 340 and 380 nm to 368 nm is not by definition linear but aerosol type related.

So in order to assess their results the authors at least have to mention the related uncertainties and the above issues raised by using retrieved AODs from two different instruments with different instrument characteristics and post processing AOD algorithms and procedures, in order to validate the Vos.

In theory a direct comparison of direct sun signals for the UVMFR instrument and a reference instrument measuring at 368nm could be used in order to assess the differences in the Vo calculation, without having the AOD calculation related uncertainties.

Please also note the supplement to this comment:
https://www.atmos-meas-tech-discuss.net/amt-2018-295/amt-2018-295-RC2-supplement.pdf

---

## Author Response (AR1)

Response to RC1

Specific Comments: In the abstract, I suggest including a sentence with the improved calibration numbers between the three methods, moving average (MA), the current operational version (OPER), and the Gaussian Process (GP) with improved dynamic input uncertainty for at least one site (HI01, IL02, or OK02).

We have revised and added the following sentences with the improved calibration numbers in the abstract (lines 23 to 29).

The validation results at the three test sites (i.e. HI02 at Mauna Loa, Hawaii, IL02 at Bondville, Illinois, and OK02 at Billings, Oklahoma) demonstrated that the agreement between aerosol optical depths (AODs) at the 368 nm channel calculated using Vo determined by the GP mean function and the equivalent AERONET AODs were consistently better than those calculated using Vo from standard techniques (e.g. moving average). For example, the average AOD biases by the GP method (0.0036 and 0.0032) are much lower than those by the moving average method (0.0119 and 0.0119) at IL02 and OK02, respectively. The GP method's absolute differences between UV-MFRSR and AERONET AOD values are approximately 4.5%, 21.6%, and 16.0% lower than those of the moving average method at HI02, IL02, and OK02, respectively.

Pg 8, line 193: Just a note to correct the wording of this sentence (though the sentence refers to irradiance at 369-nm). The Physikalish-Meteorological Observatorium Davos, World Calibration Center has a Precision Filter Radiometer (PFR) that measures AOD at 368-nm. Using this type of instrument would avoid additional uncertainties in AOD caused by the interpolation between wavelengths when comparing the MFRSR with the AERONET CIMEL. At the sites used for the comparison in this paper, the site HI02 has a PFR but I do not know about the other two sites. This isn't essential for the analysis, nor conclusions of the paper, only suggest the sentence be modified.

Thanks for pointing out the existence of the WMO reference instruments that measure at 368 nm. We have changed the beginning sentence of section 2.4 to the following (lines 199 to 205).

Ideally, to avoid additional uncertainties caused by the interpolation between wavelengths, the calibration factors should be validated via a direct comparison of direct sun signals from the to-be-calibrated UV-MFRSR and a reference instrument measuring at the 368 nm channel (e.g. the standard precision Filter radiometer (PFR) operated by the Physikalisches-Meteorologisches Observatorium Davos, World Optical Depth Research Calibration Center (WORCC)). However, such reference measurements are not available at most UVMRP stations. Therefore, the estimated mean normalized Vo (Vo_norm) values from the Gaussian Process regression and the other two comparison methods (i.e. MA and OPER) are validated indirectly in terms of aerosol optical depth (AOD) against those obtained from the collocated AERONET sites.

For validation of the technique, the authors compare AOD at 368-nm from the UVMFRSR indirectly to the AERONET CIMEL using information of AOD at two wavelengths (340 and 380 nm). Different types of measurement techniques have their own source of uncertainties as with the CIMEL and the addition of the few paragraphs on previous literature that highlights these differences is crucial to the understanding the improvements using the GP technique.

We agree that the discrepancy/uncertainties in deriving AOD values from the two instruments' measurements should be highlighted explicitly. The following sentences were added at the end of section 2.4 (lines 273 to 288).

Since AERONET and UV-MFRSR AOD values at 368 nm are derived from measurements involving different instruments and wavelengths, the uncertainties when comparing these AOD values should be noted. Some important sources of uncertainties include:

1) AERONET calibration error – At the time of calibration at MLO, AERONET reference instruments have an uncertainty of ~0.2 to 0.5%, which is equivalent to a 0.002 to 0.005 uncertainty in AERONET AOD (Holben et al. 2001). These calibration factors are likely to shift within the year following calibration, which may result in a total AOD uncertainty of ~0.01 to 0.02 (wavelength dependent, higher in the UV) (Holben et al. 2001).

2) Instrument Field of View (FOV) - AERONET CIMELs have a field-of-view (FOV) of 1.2° while the UV-MFRSR has a larger FOV (e.g. ~6.5°, reported by Kazadzis et al. 2018). AODs obtained from instruments with larger FOVs are associated with greater AOD uncertainty due to larger contributions of scattered light to the direct irradiance measurement (Kim et al. 2005).

3) Instrument maintenance – Periodic soiling and cleaning of the UV-MFRSR diffuser can result in spurious increases and decreases in AOD, respectively. The frequency of on-site maintenance (e.g. cleaning of the UV-MFRSR dome) as well as rainfall events may therefore account for some of the AOD difference (Kim et al. 2005; Kim et al. 2008).

4) Trace gases - As mentioned above, AERONET AOD accounts for $NO_2$ optical depth (e.g. ~0.002 at OK02) while UV-MFRSR AOD does not.

Pg. 7, section 2.2 on Moving Average. This doesn't describe the moving window size used in the analysis.

We have added the following sentence describing the moving window size in section 2.2 (lines 183 to 184).

The parameter *win_size* of MA is set at 20 for all applicable cases in this study.

Technical corrections:

Pg 18, line 395. AEROENT needs to be AERONET.

We have corrected the error accordingly (5 instances).

Pg 420, line 422-423. Incomplete sentence.

The sentence has been revised as below.

As a result, higher accuracy of Rayleigh and other optical depth components is required to discern small improvement on AOD for HI02.

Response to RC2

Concerning the introduction
I am missing previous results of (UV-) MFRSR comparisons with other standard AOD measuring instruments. For example in the 2015 the Filter Radiometer comparison in Davos, Switzerland various of this instrument types have participated and the results have been discussed. There are also earlier studies of such comparisons.
In Kazadzis et al., 2018: There were 4 MFRSR instruments in this campaign. The results were summarized in the following paragraph:
"*The four MFR instruments showed good agreement for the medians compared to the PFR triad, however, they exhibit larger scatter than the sun-pointing instruments resulting in a lower precision. McArthur et al. (2003) had previously reported that the MFR-derived AOD does not quite meet the accuracy of the sun-pointing instruments under clean atmospheric conditions. MFR_DE showed an AOD overestimation in various instances that gave results that are outside the WMO defined AOD limits (Fig. 2d). This small overestimation of the MFR_DE instrument compared to the PFR Triad could be due to uncertainties introduced while correcting for their angular response, the calibration procedure, or incomplete blocking of the diffuser by the shadow-band. The MFRSRs that are part of the SURFRAD network (MFR_US2 and MFR_US3) give a median AOD at 500-nm that is in very good agreement with the PFR triad and as good or better than some of the other sun-pointing instruments, e.g., CIMEL and POM; these two slightly underestimate the AOD at 865 nm, but are within the WMO defined limits. Again, these two MFRs' medians are comparable to the better sun-pointing instruments, but give larger scatter. These two MFRs are representative of the SURFRAD network that follow network protocols for calibration and alignment and frequent characterizations of the spectral and angular responses (Augustine et al., 2003, Michalsky et al., 2001).*"
Would be helpful some of the above aspects to be included in the introduction section or in section 2.4

We have added the following sentences in section 2.4 (lines 214 to 222):

During the recent Fourth Filter Radiometer Comparison held in Davos, Switzerland (between 28 September and 16 October 2015), most AOD values derived from the three AERONET CIMEL sunphotometers are within the ±0.01 range compared with the PFR triad standard (Kazadzis et al. 2018). This includes those determined at 368nm from the extrapolation of AERONET AODs at 340nm and 380nm. The 2015 Davos campaign also included four MFRSR instruments. Overall, the results showed good agreement between the four MFRSRs and the PFR triad standard, though one instrument exhibited a positive bias and low precision compared to the sun-pointing instruments (Kazadzis et al. 2018). However, such errors were likely explained by instrument-specific uncertainties (e.g. angular response correction, responsivity calibration, and shadowband position issues) and do not suggest inherent error in MFRSR AODs (Kazadzis et al. 2018).

Authors are mentioning: "There are no other instruments measuring at 368nm"
*The World Meteorological Organization (WMO) instigated the Global Atmosphere Watch (GAW)*
*program in 1989. Based on a recommendation by GAW experts, the World Optical Depth Research*
*Calibration Center (WORCC) was established in 1996 at the PMOD/WRC in Switzerland. WORCC has*
*since been advised by the GAW Scientific Advisory Group for Aerosols. The standard instrument*
*consist of a precision Filter radiometer (PFR measuring at 368, 412, 500 and 862 nm.* So actually the
WMO reference instruments (triad) is measuring at 368nm.
So the argument of the non existence of instruments measuring at 368nm (thus the choice of the
AOD based comparison) is not correct. However, as it is not possible to repeat this study with one
of the PFR instruments, probably the only solution could
be the AOD comparison with the collocated cimels as the authors have initiated. Nevertheless, a
short comment on the above text could be included in the paper.

Thanks for pointing out the existence of the WMO reference instruments that measure at 368 nm. We have changed the beginning sentence of section 2.4 to the following (lines 199 to 205).

Ideally, to avoid additional uncertainties caused by the interpolation between wavelengths, the calibration factors should be validated via a direct comparison of direct sun signals from the to-be-calibrated UV-MFRSR and a reference instrument measuring at the 368 nm channel (e.g. the standard precision Filter radiometer (PFR) operated by the Physikalisches-Meteorologisches Observatorium Davos, World Optical Depth Research Calibration Center (WORCC)). However, such reference measurements are not available at most UVMRP stations. Therefore, the estimated mean normalized Vo (Vo_norm) values from the Gaussian Process regression and the other two comparison methods (i.e. MA and OPER) are validated indirectly in terms of aerosol optical depth (AOD) against those obtained from the collocated AERONET sites.

Line 224 : probably you should comment also that ozone is also ignored .

We changed the word "ozone" to "$O_3$" to make it clear ozone is also ignored.

Line 235: I guess that the cloud flagging method is not evaluated here, as comparison with CIMEL data includes only data that CIMEL algorithm considers as cloud free.

Yes, we did not evaluate the cloud flagging method in this study. We mainly relied on AERONET's cloud flagging for selecting cloud-free UV-MFRSR measurements. We did perform a simple variation check using similar methodologies to those found in many cloud screening algorithms (e.g. Alexandrov et al. 2004) to reduce the potential contamination of aerosol optical depth by broken clouds.

Line 250 : It would be informative to explain why S(lamda) appears in equation 9.

Since F(lamda) is ~4nm the integrated range is ~366-370 nm. There you mention that AOD is the "interpolated ADO spectrum" which I guess you mean the linear (?) interpolated using 340 and 380 AERONET AODs ? Then S(lamda) is used for normalization in this small 4 nm range? Is this so different that the actual interpolated value of AOD at 368 nm? And if S(lamda) is used, why not S(lamda) – Rayleigh optical depth ? Have in mind that spectral function FWHM of the CIMEL is larger than 2nm.

We agree that the S(lambda) in equation 9 has minimal effects on deriving the UVRSR 368 band passed AERONET AOD. For example, for Mauna Loa, Hawaii site, the mean ($2.6 \times 10^{-6}$) and standard deviation ($2.5 \times 10^{-6}$) of the difference between the AOD with and without S(lambda) are several magnitudes smaller than the instrument resolution. Therefore, we removed the S(lambda) term in equation 9 for simplicity (lines 266 to 271). The change is so small that it doesn't impact further analysis in the manuscript.

Comparing the AOD directly interpolated to 368 nm with the bandpass AOD (calculated with updated equation 9) for Mauna Loa, Hawaii site, we found that there was a ~0.1% discrepancy. The small discrepancy could be explained by the small difference between the effective wavelength of the instrument (367.91 nm for the MLO UV-MFRSR) and 368 nm. With larger wavelength differences for some instruments, we expect a larger discrepancy. Therefore, we decided to keep using the more accurate bandpass AOD in the manuscript.

The authors chose to evaluate their method by comparing the retrieved (from their Vos data) AODs. Here are my comments on this section:

- The comparison of UVMFR with AERONET would be essential to follow criteria that are defined by WMO –CIMO in order to assess the results in detail.

https://library.wmo.int/pmb_ged/wmo-td_1287.pdf

There (page 8) such conditions and formulas are defined.

For example the U95 criterion where a number (here the lower limit is 95%) of measurements have to be in the range of ±0.005 + 0.010/m

Where 0.005 accounts for instrument related uncertainties and 0.01/m for calibration related uncertainties (calibration uncertainty better than 1%).

Changing the analysis figures with the use of this criterion authors can:

a. better show the agreement and the improvements with their methods by showing the percentage of data within these limits for each case.

b. having in mind that calibration related uncertainties will be inherited in AOD retrievals as a function of air mass, the figures (a) including aod differences vs air mass can point out on Vo related issues.

Still slopes and cor. Coefficients can be reported in the form of a table.

Thank you for pointing out the U95 criterion. We understand that complying with such criterion is critical for achieving traceability of AOD products generated from radiometer measurements.

In this study, the comparison of AOD values derived from UV-MFRSR measurements and AERONET AOD values only serves as an indirect evidence that the calibration of UV-MFRSR is reasonably accurate. However, we do not attempt to argue that the calibration of UVMRP UV-MFRSR is accurate enough to produce AOD values that meet the U95 criterion. Nor do we attempt to argue that UVMRP AOD products are traceable to the AERONET or WMO AOD standard. As mentioned in the manuscript and the citations therein, the stability assumption of the Langley method may not be strictly fulfilled at many UVMRP sites, rendering larger uncertainty of Langley calibration factors at these sites compared with those derived under ideal conditions (such as at Mauna Loa, Hawaii).

This conclusion is supported by the following analysis. Based on the GP uncertainty results (using 1.96 instead of 4.42 standard deviations to mimic 95% CI, data not shown in the manuscript) among the three test sites, only at HI02 (Mauna Loa, Hawaii), the AOD uncertainty (at 95% level) caused by calibration (~0.0078) is lower than the U95 criterion (0.01). The AOD uncertainties (at 95% level) caused by calibration at the other two sites are much larger than 0.01 (i.e. ~0.043 for IL02, and ~0.028 for OK02).

We recognize that the assumption that AERONET AOD represents "ground-truth" is not ideal, however, as indicated by previous work, it is a reasonable assumption for this study. AERONET sunphotometers are routinely calibrated with the uncertainty of AOD around 0.002 to 0.005 in the visible and up to 0.01 in the UV region (Eck et al., 1999; Holben et al., 2001). Additionally, during the recent Fourth Filter

Radiometer Comparison held in Davos, Switzerland (between 28 September and 16 October 2015), most AOD values derived from the three AERONET CIMEL sunphotometers are within the ±0.01 range compared with the PFR triad (Kazadzis et al. 2018). This includes those determined at 368nm from the extrapolation of AERONET AODs at 340nm and 380nm. Therefore, we are confident that the AERONET AOD products used in this study are accurate enough to be a reliable source of AOD values for validation purposes. This assumption is consistent with numerous other field-based evaluations of radiometric AOD accuracy as detailed between lines 213 and 214 in the manuscript. To emphasize the validity of using AERONET as an effective standard, we have revised the following text in the manuscript summarizing the results from the Fourth Filter Radiometer Comparison and Kazadzis et al. (2018) (lines 199 to 222).

Ideally, to avoid additional uncertainties caused by the interpolation between wavelengths, the calibration factors should be validated via a direct comparison of direct sun signals from the to-be-calibrated UV-MFRSR and a reference instrument measuring at the 368 nm channel (e.g. the standard precision Filter radiometer (PFR) operated by the Physikalisches-Meteorologisches Observatorium Davos, World Optical Depth Research Calibration Center (WORCC)). However, such reference measurements are not available at most UVMRP stations. Therefore, the estimated mean normalized Vo (Vo_norm) values from the Gaussian Process regression and the other two comparison methods (i.e. MA and OPER) are validated indirectly in terms of aerosol optical depth (AOD) against those obtained at the collocated AERONET sites. We admit that the uncertainty of UV-MFRSR AODs could exceed the World Meteorological Organization (WMO) U95 criterion (e.g. 95% of the measured data have uncertainty in the range of 0.005 ± 0.01 / airmass, Kazadzis et al. 2018) at many UVMRP sites because the stability assumption of the Langley method may not be strictly fulfilled. Therefore, the AOD comparison in this study can only serve as an indirect evidence to verify whether the calibration of UV-MFRSR is reasonably accurate.

AERONET sunphotometers are routinely calibrated with the uncertainty of AOD around 0.002 to 0.005 in the visible and up to 0.01 in the UV region (Eck et al., 1999; Holben et al., 2001) and are therefore considered a reliable source for AOD intercomparison and radiometer validation [e.g. (Alexandrov et al., 2002, 2008;Augustine et al., 2003;Krotkov et al., 2005a;Krotkov et al., 2005b;Kassianov et al., 2007;Tang et al., 2013;Yin et al., 2015;Zhang et al., 2016)]. During the recent Fourth Filter Radiometer Comparison held in Davos, Switzerland (between 28 September and 16 October 2015), most AOD values derived from the three AERONET CIMEL sunphotometers are within the ±0.01 range compared with the PFR triad standard (Kazadzis et al. 2018). This includes those determined at 368nm from the extrapolation of AERONET AODs at 340nm and 380nm. The 2015 Davos campaign also included four MFRSR instruments. Overall, the results showed good agreement between the four MFRSRs and the PFR triad standard, though one instrument exhibited a positive bias and low precision compared to the sun-pointing instruments (Kazadzis et al. 2018). However, such errors were likely explained by instrument-specific uncertainties (e.g. angular response correction, responsivity calibration, and shadowband position issues) and do not suggest inherent error in MFRSR AODs (Kazadzis et al. 2018).

The AOD retrieval and the differences among two instruments are a consequence not only an uncertainty on the instrument calibrations but also other factors.
Here is a list:
- The calculation of Rayleigh optical depth from both instruments including the pressure measurement. Are the two instruments ( UVMFR and CIMEL) use the same formulas ?
- The calculation of Rayleigh and aerosol air mass factors
- The potential differences in the field of view of the instrument
- CIMEL includes NO2 and Ozone optical depths
- The wavelength interpolation from 340 and 380 nm to 368 nm is not by definition linear but aerosol type related.
So in order to assess their results the authors at least have to mention the related uncertainties and the above issues raised by using retrieved AODs from two different instruments with different instrument characteristics and post processing AOD algorithms and procedures, in order to validate the Vos.
In theory a direct comparison of direct sun signals for the UVMFR instrument and a reference instrument measuring at 368nm could be used in order to assess the differences in the Vo calculation, without having the AOD calculation related uncertainties.

We agree that the discrepancy/uncertainties in deriving AOD values from the two instruments' measurements should be highlighted explicitly. The following sentences were added at the end of section 2.4 (lines 273 to 288).

Since AERONET and UV-MFRSR AOD values at 368 nm are derived from measurements involving different instruments and wavelengths, the uncertainties when comparing these AOD values should be noted. Some important sources of uncertainties include:

1) AERONET calibration error – At the time of calibration at MLO, AERONET reference instruments have an uncertainty of ~0.2 to 0.5%, which is equivalent to a 0.002 to 0.005 uncertainty in AERONET AOD (Holben et al. 2001). These calibration factors are likely to shift within the year following calibration, which may result in a total AOD uncertainty of ~0.01 to 0.02 (wavelength dependent, higher in the UV) (Holben et al. 2001).
2) Instrument Field of View (FOV) - AERONET CIMELs have a field-of-view (FOV) of 1.2° while the UV-MFRSR has a larger FOV (e.g. ~6.5°, reported by Kazadzis et al. 2018). AODs obtained from instruments with larger FOVs are associated with greater AOD uncertainty due to larger contributions of scattered light to the direct irradiance measurement (Kim et al. 2005).
3) Instrument maintenance – Periodic soiling and cleaning of the UV-MFRSR diffuser can result in spurious increases and decreases in AOD, respectively. The frequency of on-site maintenance (e.g. cleaning of the UV-MFRSR dome) as well as rainfall events may therefore account for some of the AOD difference (Kim et al. 2005; Kim et al. 2008).
4) Trace gases - As mentioned above, AERONET AOD accounts for $NO_2$ optical depth (e.g. ~0.002 at OK02) while UV-MFRSR AOD does not.

We do not include the Rayleigh optical depth formula, airmass formula, and interpolation methods in the list.

For the Rayleigh optical depth (RLOD) calculation, both UV-MFRSR and AERONET (version 2, https://aeronet.gsfc.nasa.gov/version2_table.pdf) use the same formula described in Bodhaine et al. 1999. The instantaneous pressure values for UV-MFRSR are obtained from the collocated AERONET measurements. Therefore, RLOD should not introduce additional uncertainty in this study.

For the airmass calculation, both UV-MFRSR and AERONET (version 2, https://aeronet.gsfc.nasa.gov/version2_table.pdf) use the same formula described in Kasten and Young 1989.

For the interpolation between 340 and 380 nm AOD values, we agree that the spectra of aerosol optical depth is aerosol type related and may not be strictly linear (e.g. slightly quadratic). However, we believe that the difference in the interpolated AOD spectrum among different interpolation methods/equations should be negligible because the two wavelengths are so close (e.g. Figure 6 in Krotkov et al. 2005a).

A list of all relevant changes made in the manuscript

1. In the abstract, we added a few sentences with the improved calibration numbers.
2. In section 2.2 Moving Average (MA), we added one sentence describing the win_size of MA used in this study.
3. In section 2.4 Validation method for 368-nm in-situ calibration factors,
    a. We correct our previous statement regarding the existence of other instruments measuring at 368 nm (i.e. the PFR operated by WORCC).
    b. We discussed the WMO U95 criterion and explained why UVMRP UV-MFRSR AODs could exceed the criterion.
    c. We added evidence from the recent Fourth Filter Radiometer Comparison (Davos, Switzerland) in 2005 to show the good agreement among AERONET CIMEL, MFRSR, and the PRF triad standard AOD measurements.
    d. We simplified the equation 9 for the equivalent AERONET AOD at the 368 nm channel.
    e. We added a paragraph discussing the potential uncertainties when comparing AERONET and UV-MFRSR AOD values at 368 nm.

[revised manuscript text omitted]

**2.4 Validation method for 368-nm in-situ calibration factors**

~~Since there are no other ground radiometers that measure the irradiance at the exact 368 nm channel as the UVMRP UV-MFRSR does, the estimated mean normalized Vo (Vo_norm) values from the Gaussian Process regression and the other two comparison methods (i.e. MA and OPER) are validated indirectly in terms of aerosol optical depth (AOD) against those obtained at the collocated AERONET sites. AERONET sunphotometers are routinely calibrated with the uncertainty of AOD around 0.002 to 0.005 in the visible and up to 0.01 in the UV region (Eck et al., 1999;Holben et al., 2001) and are therefore considered a reliable source for AOD intercomparison and radiometer validation [e.g. (Alexandrov et al., 2002, 2008;Augustine et al., 2003;Krotkov et al., 2005a;Krotkov et al., 2005b;Kassianov et al., 2007;Tang et al., 2013;Yin et al., 2015;Zhang et al., 2016)].~~ Ideally, to avoid additional uncertainties caused by the interpolation between wavelengths, the calibration factors should be validated via a direct comparison of direct sun signals from the to-be-calibrated UV-MFRSR and a reference instrument measuring at the 368 nm channel [e.g. the standard precision Filter radiometer (PFR) operated by the Physikalisches-Meteorologisches Observatorium Davos, World Optical Depth Research Calibration Center (WORCC)]. However, such reference measurements are not available at most UVMRP stations. Therefore, the estimated mean normalized Vo (Vo_norm) values from the Gaussian Process regression and the other two comparison methods (i.e. MA and OPER) are validated indirectly in terms of aerosol optical depth (AOD) against those obtained at the collocated AERONET sites. We admit that the uncertainty of UV-MFRSR AODs could exceed the World Meteorological Organization (WMO) U95 criterion [e.g. 95% of the measured data have uncertainty in the range of $0.005 \pm 0.01$ / airmass, (Kazadzis et al., 2018)] at many UVMRP sites because the stability assumption of the Langley method may not be strictly fulfilled. Therefore, the AOD comparison in this study can only serves as an indirect evidence to verify whether the calibration of UV-MFRSR is reasonably accurate.

AERONET sunphotometers are routinely calibrated with the uncertainty of AOD around 0.002 to 0.005 in the visible and up to 0.01 in the UV region (Eck et al., 1999;Holben et al., 2001) and are therefore considered a reliable source for AOD intercomparison and radiometer validation [e.g. (Alexandrov et al., 2002, 2008;Augustine et al., 2003;Krotkov et al., 2005a;Krotkov et al., 2005b;Kassianov et al., 2007;Tang et al., 2013;Yin et al., 2015;Zhang et al., 2016)]. During the recent Fourth Filter Radiometer Comparison held in Davos, Switzerland (between 28 September and 16 October 2015), most AOD

values derived from the three AERONET CIMEL sunphotometers are within the ±0.01 range compared with the PFR triad standard (Kazadzis et al., 2018). This includes those determined at 368nm from the extrapolation of AERONET AODs at 340nm and 380nm. The 2015 Davos campaign also included four MFRSR instruments. Overall, the results showed good agreement between the four MFRSRs and the PFR triad standard, though one instrument exhibited a positive bias and low precision compared to the sun-pointing instruments (Kazadzis et al., 2018). However, such errors were likely explained by instrument-specific uncertainties (e.g. angular response correction, responsivity calibration, and shadowband position issues) and do not suggest inherent error in MFRSR AODs (Kazadzis et al., 2018).

[revised manuscript text omitted]

AERONET (v2.0) provides AOD at 340 and 380 nm channels. These values are interpolated to the effective wavelength of the UV-MFRSR 368 nm channel for comparison using the Ångström exponent as follows. Note that in the log transformed coordinate system [i.e. log(AOD) vs. log(wavelength)], log(AOD) is generally linear between 340 and 380 nm (Krotkov et al., 2005a). First, the AERONET AOD spectrum between the two wavelengths is derived by linear interpolation of AERONET AODs at 340 and 380 nm in the log transformed coordinate system. Next, since the UV-MFRSR AOD at 368 nm is a bandpass value over a narrow band (i.e 2 nm FHMW), the equivalent AERONET AOD at that channel is derived by

$$AOD_{368nm,AERONET} = \frac{\int_{340nm}^{380nm} AOD_{\lambda} F_{\lambda} d\lambda}{\int_{340nm}^{380nm} F_{\lambda} d\lambda} \, , \qquad (9)$$

where $AOD_{\lambda}$ is the interpolated AERONET AOD spectrum; $F_{\lambda}$ is the spectral response function of the UV-MFRSR at 368 nm channel (http://uvb.nrel.colostate.edu/UVB/da_queryFilterFunctions.jsf);  and the wavelength interval for the integral is 0.05 nm. Note that negative AERONET AOD measurements are excluded from the validation because of using log transform.

Since AERONET and UV-MFRSR AOD values at 368 nm are derived from measurements involving different instruments and wavelengths, the uncertainties when comparing these AOD values should be noted. Some important sources of uncertainties include:

1) AERONET calibration error – At the time of calibration at MLO, AERONET reference instruments have an uncertainty of ~0.2 to 0.5%, which is equivalent to a 0.002 to 0.005 uncertainty in AERONET AOD (Holben et al.,

2001). These calibration factors are likely to shift within the year following calibration, which may result in a total AOD uncertainty of ~0.01 to 0.02 (wavelength dependent, higher in the UV) (Holben et al., 2001).

2) Instrument Field of View (FOV) – AERONET CIMELs have a FOV of 1.2° while the UV-MFRSR has a larger FOV
295 [e.g. ~6.5°, reported by (Kazadzis et al., 2018)]. AODs obtained from instruments with larger FOVs are associated with greater AOD uncertainty due to larger contributions of scattered light to the direct irradiance measurement (Kim et al., 2005).

3) Instrument maintenance – Periodic soiling and cleaning of the UV-MFRSR diffuser can result in spurious increases and decreases in AOD, respectively. The frequency of on-site maintenance (e.g. cleaning of the UV-MFRSR dome)
300 as well as rainfall events may therefore account for some of the AOD difference (Kim et al., 2005;Kim et al., 2008).

4) Trace gases – As mentioned above, AERONET AOD accounts for $NO_2$ optical depth (e.g. ~0.002-0.003 at OK02) while UV-MFRSR AOD does not.

[revised manuscript text omitted]

| | $\mathrm{Avg}\left(\dfrac{\lvert\mathrm{AOD}_{368,UVMRP}-\mathrm{AOD}_{368,AE}\rvert}{\mathrm{AOD}_{368,AE}}\right)$ | 0.5803 | 0.6078 | 0.6261 |
| | LR | y=1.0550x+0.0045 | y=1.0551x+0.0047 | y=1.0601x+0.0043 |
| | $R^2$ | 0.9000 | 0.8957 | 0.8812 |
| IL02 | Avg($\lvert$AOD$_{368,UVMRP}$-AOD$_{368,AE}\rvert$) | 0.0228 | 0.0291 | 0.0270 |
| | $\mathrm{Avg}\left(\dfrac{\lvert\mathrm{AOD}_{368,UVMRP}-\mathrm{AOD}_{368,AE}\rvert}{\mathrm{AOD}_{368,AE}}\right)$ | 0.1669 | 0.2087 | 0.1930 |
| | LR | y=0.9615x+0.0115 | y=0.9543x+0.0213 | y=0.9241x+0.0065 |
| | $R^2$ | 0.9514 | 0.9420 | 0.9332 |
| OK02 | Avg($\lvert$AOD$_{368,UVMRP}$-AOD$_{368,AE}\rvert$) | 0.0150 | 0.01785 | 0.01847 |
| | $\mathrm{Avg}\left(\dfrac{\lvert\mathrm{AOD}_{368,UVMRP}-\mathrm{AOD}_{368,AE}\rvert}{\mathrm{
[revised manuscript text omitted]

Kazadzis, S., Kouremeti, N., Diémoz, H., Gröbner, J., Forgan, B. W., Campanelli, M., Estellés, V., Lantz, K., Michalsky, J.,

630      Carlund, T., Cuevas, E., Toledano, C., Becker, R., Nyeki, S., Kosmopoulos, P. G., Tatsiankou, V., Vuilleumier, L., Denn, F. M., Ohkawara, N., Ijima, O., Goloub, P., Raptis, P. I., Milner, M., Behrens, K., Barreto, A., Martucci, G., Hall, E., Wendell, J., Fabbri, B. E., and Wehrli, C.: Results from the Fourth WMO Filter Radiometer Comparison for aerosol optical depth measurements, Atmos. Chem. Phys., 18, 3185-3201, 10.5194/acp-18-3185-2018, 2018.

Kim, S.-W., Jefferson, A., Yoon, S.-C., Dutton, E. G., Ogren, J. A., Valero, F. P. J., Kim, J., and Holben, B. N.: Comparisons

635      of aerosol optical depth and surface shortwave irradiance and their effect on the aerosol surface radiative forcing estimation, Journal of Geophysical Research: Atmospheres, 110, doi:10.1029/2004JD004989, 2005.

Kim, S.-W., Yoon, S.-C., Dutton, E. G., Kim, J., Wehrli, C., and Holben, B. N.: Global Surface-Based Sun Photometer Network for Long-Term Observations of Column Aerosol Optical Properties: Intercomparison of Aerosol Optical Depth, Aerosol Science and Technology, 42, 1-9, 10.1080/02786820701699743, 2008.

640 Krotkov, N. A., Bhartia, P. K., Herman, J. R., Slusser, J. R., Labow, G. J., Scott, G. R., Janson, G. T., Eck, T., and Holben, B. N.: Aerosol ultraviolet absorption experiment (2002 to 2004), part 1: ultraviolet multifilter rotating shadowband radiometer calibration and intercomparison with CIMEL sunphotometers, Optical Engineering, 44, 041004, 10.1117/1.1886818, 2005a.

Krotkov, N. A., Bhartia, P. K., Herman, J. R., Slusser, J. R., Scott, G. R., Labow, G. J., Vasilkov, A. P., Eck, T., Doubovik, O., and Holben, B. N.: Aerosol ultraviolet absorption experiment (2002 to 2004), part 2: absorption optical thickness, refractive

645 index, and single scattering albedo, Optical Engineering, 44, 041005-041001–041005-041017, 2005b.

Kupilik, M., and Witmer, F.: Spatio-temporal violent event prediction using Gaussian process regression, Journal of Computational Social Science, 10.1007/s42001-018-0024-y, 2018.

Mather, J. H., and Voyles, J. W.: The Arm Climate Research Facility: A Review of Structure and Capabilities, Bulletin of the American Meteorological Society, 94, 377-392, 10.1175/bams-d-11-00218.1, 2013.

650 Press, W. H.: Numerical recipes in C : the art of scientific computing, 2nd ed., Cambridge University Press, Cambridge Cambridgeshire, New York, 1992.

Proietti, T.: Trend Estimation, in: International Encyclopedia of Statistical Science, edited by: Lovric, M., Springer Berlin Heidelberg, Berlin, Heidelberg, 1613-1616, 2011.

Rasmussen, C. E., and Williams, C. K. I.: Gaussian processes for machine learning, Cambridge, MA: MIT Press, Cambridge,

655 MA, USA, 266 pp., 2006.

Richter, P., and Toledano-Ayala, M.: Revisiting Gaussian Process Regression Modeling for Localization in Wireless Sensor Networks, Sensors, 15, 22587, 2015.

Shaw, G. E.: Error analysis of multi-wavelength sun photometry, pure and applied geophysics, 114, 1-14, 10.1007/bf00875487, 1976.

660 Slusser, J., Gibson, J., Bigelow, D., Kolinski, D., Disterhoft, P., Lantz, K., and Beaubien, A.: Langley method of calibrating UV filter radiometers, Journal of Geophysical Research: Atmospheres, 105, 4841-4849, 10.1029/1999JD900451, 2000.

Tang, H., Chen, M., Davis, J., and Gao, W.: Comparison of aerosol optical depth of UV-B monitoring and research program (UVMRP), AERONET and MODIS over continental united states, Frontiers of Earth Science, 7, 129-140, 10.1007/s11707-013-0376-9, 2013.

665 Viereck, R. A., Floyd, L. E., Crane, P. C., Woods, T. N., Knapp, B. G., Rottman, G., Weber, M., Puga, L. C., and DeLand, M. T.: A composite Mg II index spanning from 1978 to 2003, Space Weather, 2, 1-13, 10.1029/2004SW000084, 2004.

Wahba, G.: Smoothing Splines, in: International Encyclopedia of Statistical Science, edited by: Lovric, M., Springer Berlin Heidelberg, Berlin, Heidelberg, 1349-1353, 2011.

Wu, R., and Wang, B.: Gaussian process regression method for forecasting of mortality rates, Neurocomputing,

670 10.1016/j.neucom.2018.08.001, 2018.

Yin, B., Min, Q., and Joseph, E.: Retrievals and uncertainty analysis of aerosol single scattering albedo from MFRSR measurements, Journal of Quantitative Spectroscopy and Radiative Transfer, 150, 95-106, 10.1016/j.jqsrt.2014.08.012, 2015.

Zhang, M., Gong, W., Ma, Y., Wang, L., and Chen, Z.: Transmission and division of total optical depth method: A universal calibration method for Sun photometric measurements, Geophysical Research Letters, 43, 2974-2980, 10.1002/2016GL068031, 2016.

675

---

## Author Response (AR2)

Response to Editor

Dear Andrew,

Thank you and the reviewer for a quick review and for providing valuable suggestions. I apologize for the slight delay of my response. We spent some time trying different software (e.g. Matlab and Excel) and finally generated the suggested plot (i.e. boxplots with addition lines) in Python.

We have added the suggested plot and a few sentences describing it in Appendix C. We introduced them after presenting the analysis of Figure 4 (lines 437-438).
* * *
**Appendix C. Comparison of UVMRP and AERONET 368-nm AOD differences as a function of airmass among the three methods.**

[Figure]

**Figure C1. Box and whisker plots of AOD$_{368,UVMRP}$-AOD$_{368,AE}$ as a function of airmass by the three methods (i.e. from top to bottom: GP, MA, OPER) at the three test sites (i.e. from left to right: HI02, IL02, OK02). Each blue bin covers a 0.25 airmass range. The red dashed lines show the WMO AOD U95 upper and lower limits.**

Figure C1 showed that GP had narrower error ranges compared with the other two methods (i.e. MA and OPER) at all three test sites (i.e. HI02, IL02, and OK02). The median values (the black short lines in blue boxes) of GP are closer to zero at IL02 and OK02 sites, especially for lower airmasses. However, regardless of site, airmass, and method, the difference between AERONET and UV-MFRSR AODs still exceeds the WMO AOD U95 criterion for a number of instances.

I hope the new figure and text have addressed the weakness of the previous manuscript. Please let us know if any modification is needed.

Best regards,

Maosi

List of all relevant changes made in the manuscript

1. Add box and whisker plots showing the UVMRP and AERONET 368 nm AOD difference as a function of airmass in Appendix C.
2. Add a sentence about Appendix C in section 2.4.
3. Add a sentence about Appendix C in section 3.2.2.

[revised manuscript text omitted]

AERONET (v2.0) provides AOD at 340 and 380 nm channels. These values are interpolated to the effective wavelength of the UV-MFRSR 368 nm channel for comparison using the Ångström exponent as follows. Note that in the log transformed coordinate system [i.e. log(AOD) vs. log(wavelength)], log(AOD) is generally linear between 340 and 380 nm (Krotkov et al., 2005a). First, the AERONET AOD spectrum between the two wavelengths is derived by linear interpolation of AERONET AODs at 340 and 380 nm in the log transformed coordinate system. Next, since the UV-MFRSR AOD at 368 nm is a bandpass value over a narrow band (i.e 2 nm FHMW), the equivalent AERONET AOD at that channel is derived by

$$AOD_{368nm,AERONET} = \frac{\int_{340nm}^{380nm} AOD_\lambda F_\lambda d\lambda}{\int_{340nm}^{380nm} F_\lambda d\lambda} \; , \tag{9}$$

where $AOD_\lambda$ is the interpolated AERONET AOD spectrum; $F_\lambda$ is the spectral response function of the UV-MFRSR at 368 nm channel (http://uvb.nrel.colostate.edu/UVB/da_queryFilterFunctions.jsf); and the wavelength interval for the integral is 0.05 nm. Note that negative AERONET AOD measurements are excluded from the validation because of using log transform.

Since AERONET and UV-MFRSR AOD values at 368 nm are derived from measurements involving different instruments and wavelengths, the uncertainties when comparing these AOD values should be noted. Some important sources of uncertainties include:

1) AERONET calibration error – At the time of calibration at MLO, AERONET reference instruments have an uncertainty of ~0.2 to 0.5%, which is equivalent to a 0.002 to 0.005 uncertainty in AERONET AOD (Holben et al., 2001). These calibration factors are likely to shift within the year following calibration, which may result in a total AOD uncertainty of ~0.01 to 0.02 (wavelength dependent, higher in the UV) (Holben et al., 2001).

2) Instrument Field of View (FOV) – AERONET CIMELs have a FOV of 1.2° while the UV-MFRSR has a larger FOV [e.g. ~6.5°, reported by (Kazadzis et al., 2018)]. AODs obtained from instruments with larger FOVs are associated with greater AOD uncertainty due to larger contributions of scattered light to the direct irradiance measurement (Kim et al., 2005).

3) Instrument maintenance – Periodic soiling and cleaning of the UV-MFRSR diffuser can result in spurious increases and decreases in AOD, respectively. The frequency of on-site maintenance (e.g. cleaning of the UV-MFRSR dome) as well as rainfall events may therefore account for some of the AOD difference (Kim et al., 2005;Kim et al., 2008).

4) Trace gases – As mentioned above, AERONET AOD accounts for $NO_2$ optical depth (e.g. ~0.002-0.003 at OK02) while UV-MFRSR AOD does not.

[revised manuscript text omitted]

**Appendix C. Comparison of UVMRP and AERONET 368-nm AOD differences as a function of airmass among the**
**three methods.**

[Figure]

**Figure C1. Box and whisker plots of AOD$_{368,UVMRP}$-AOD$_{368,AE}$ as a function of airmass by the three methods (i.e. from top to bottom: GP, MA, OPER) at the three test sites (i.e. from left to right: HI02, IL02, OK02). Each blue bin covers a 0.25 airmass range. The red dashed lines show the WMO AOD U95 upper and lower limits.**

Figure C1 showed that GP had narrower error ranges compared with the other two methods (i.e. MA and OPER) at all three test sites (i.e. HI02, IL02, and OK02). The median values (the black short lines in blue boxes) of GP are closer to zero at IL02 and OK02 sites, especially for lower airmasses. However, regardless of site, airmass, and method, the difference between AERONET and UV-MFRSR AODs still exceeds the WMO AOD U95 criterion for a number of instances.

[revised manuscript text omitted]

Kazadzis, S., Kouremeti, N., Diémoz, H., Gröbner, J., Forgan, B. W., Campanelli, M., Estellés, V., Lantz, K., Michalsky, J.,

Carlund, T., Cuevas, E., Toledano, C., Becker, R., Nyeki, S., Kosmopoulos, P. G., Tatsiankou, V., Vuilleumier, L., Denn, F. M., Ohkawara, N., Ijima, O., Goloub, P., Raptis, P. I., Milner, M., Behrens, K., Barreto, A., Martucci, G., Hall, E., Wendell, J., Fabbri, B. E., and Wehrli, C.: Results from the Fourth WMO Filter Radiometer Comparison for aerosol optical depth measurements, Atmos. Chem. Phys., 18, 3185-3201, 10.5194/acp-18-3185-2018, 2018.

Kim, S.-W., Jefferson, A., Yoon, S.-C., Dutton, E. G., Ogren, J. A., Valero, F. P. J., Kim, J., and Holben, B. N.: Comparisons of aerosol optical depth and surface shortwave irradiance and their effect on the aerosol surface radiative forcing estimation, Journal of Geophysical Research: Atmospheres, 110, doi:10.1029/2004JD004989, 2005.

Kim, S.-W., Yoon, S.-C., Dutton, E. G., Kim, J., Wehrli, C., and Holben, B. N.: Global Surface-Based Sun Photometer Network for Long-Term Observations of Column Aerosol Optical Properties: Intercomparison of Aerosol Optical Depth, Aerosol Science and Technology, 42, 1-9, 10.1080/02786820701699743, 2008.

Krotkov, N. A., Bhartia, P. K., Herman, J. R., Slusser, J. R., Labow, G. J., Scott, G. R., Janson, G. T., Eck, T., and Holben, B. N.: Aerosol ultraviolet absorption experiment (2002 to 2004), part 1: ultraviolet multifilter rotating shadowband radiometer calibration and intercomparison with CIMEL sunphotometers, Optical Engineering, 44, 041004, 10.1117/1.1886818, 2005a.

Krotkov, N. A., Bhartia, P. K., Herman, J. R., Slusser, J. R., Scott, G. R., Labow, G. J., Vasilkov, A. P., Eck, T., Doubovik, O., and Holben, B. N.: Aerosol ultraviolet absorption experiment (2002 to 2004), part 2: absorption optical thickness, refractive index, and single scattering albedo, Optical Engineering, 44, 041005-041001–041005-041017, 2005b.

Kupilik, M., and Witmer, F.: Spatio-temporal violent event prediction using Gaussian process regression, Journal of Computational Social Science, 10.1007/s42001-018-0024-y, 2018.

Mather, J. H., and Voyles, J. W.: The Arm Climate Research Facility: A Review of Structure and Capabilities, Bulletin of the American Meteorological Society, 94, 377-392, 10.1175/bams-d-11-00218.1, 2013.

Press, W. H.: Numerical recipes in C : the art of scientific computing, 2nd ed., Cambridge University Press, Cambridge Cambridgeshire, New York, 1992.

Proietti, T.: Trend Estimation, in: International Encyclopedia of Statistical Science, edited by: Lovric, M., Springer Berlin Heidelberg, Berlin, Heidelberg, 1613-1616, 2011.

Rasmussen, C. E., and Williams, C. K. I.: Gaussian processes for machine learning, Cambridge, MA: MIT Press, Cambridge,

MA, USA, 266 pp., 2006.

Richter, P., and Toledano-Ayala, M.: Revisiting Gaussian Process Regression Modeling for Localization in Wireless Sensor Networks, Sensors, 15, 22587, 2015.

Shaw, G. E.: Error analysis of multi-wavelength sun photometry, pure and applied geophysics, 114, 1-14, 10.1007/bf00875487, 1976.

Slusser, J., Gibson, J., Bigelow, D., Kolinski, D., Disterhoft, P., Lantz, K., and Beaubien, A.: Langley method of calibrating UV filter radiometers, Journal of Geophysical Research: Atmospheres, 105, 4841-4849, 10.1029/1999JD900451, 2000.

Tang, H., Chen, M., Davis, J., and Gao, W.: Comparison of aerosol optical depth of UV-B monitoring and research program (UVMRP), AERONET and MODIS over continental united states, Frontiers of Earth Science, 7, 129-140, 10.1007/s11707-013-0376-9, 2013.

Viereck, R. A., Floyd, L. E., Crane, P. C., Woods, T. N., Knapp, B. G., Rottman, G., Weber, M., Puga, L. C., and DeLand, M. T.: A composite Mg II index spanning from 1978 to 2003, Space Weather, 2, 1-13, 10.1029/2004SW000084, 2004.

Wahba, G.: Smoothing Splines, in: International Encyclopedia of Statistical Science, edited by: Lovric, M., Springer Berlin Heidelberg, Berlin, Heidelberg, 1349-1353, 2011.

Wu, R., and Wang, B.: Gaussian process regression method for forecasting of mortality rates, Neurocomputing,

10.1016/j.neucom.2018.08.001, 2018.

Yin, B., Min, Q., and Joseph, E.: Retrievals and uncertainty analysis of aerosol single scattering albedo from MFRSR measurements, Journal of Quantitative Spectroscopy and Radiative Transfer, 150, 95-106, https://doi.org/10.1016/j.jqsrt.2014.08.012, 2015.

Zhang, M., Gong, W., Ma, Y., Wang, L., and Chen, Z.: Transmission and division of total optical depth method: A universal calibration method for Sun photometric measurements, Geophysical Research Letters, 43, 2974-2980, 10.1002/2016GL068031, 2016.